# M-IDoL: Information Decomposition for Modality-Specific and Diverse Representation Learning in Medical Foundation Model

**Yihang Liu**[1]  **Longzhen Yang**[1]  **Jiaxiong Yang**[1]  **Ying Wen**[2]  **Lianghua He**[1]  **Heng Tao Shen**[1]

## Abstract

Medical foundation models (MFMs) aim to learn universal representations from multimodal medical images that can generalize effectively to diverse downstream clinical tasks. However, most existing MFMs suffer from information ambiguity that blends multimodal representations in a single embedding space, leading to the degradation of modality specificity and diversity. In this paper, we propose M-IDoL, a self-supervised _M_FM that introduces _I_nformation _D_ecomposition for multim_o_dal representation _L_earning via two objectives: i) maximizing inter-modality entropy by dispersing multimodal representations into separable Mixture-of-Experts (MoE) subspaces to achieve representation specificity across modalities; and ii) minimizing intra-modality uncertainty by performing fine-grained semantic discrimination within each MoE subspace to enrich representation diversity per modality. By pre-training on 1.15 million medical images, M-IDoL i) delivers superior generalization across 21 downstream clinical tasks, outperforming 20 foundation models on five imaging modalities (e.g., X-ray, fundus, OCT, dermoscopy and pathology), and ii) learns modality-specific and diverse representations, showing clearer separation of feature clusters across modalities and finer-grained feature discrimination within each modality.

## 1. Introduction

Medical foundation models (MFMs) have emerged as a promising paradigm for multi-domain medical imaging (Ye

[1]School of Computer Science and Technique, Tongji University, Shanghai 201804, China. [2]School of Communications and Electronic Engineering, East China Normal University, Shanghai 200241, China.. Correspondence to: Longzhen Yang <yanglongzhen@tongji.edu.cn>, Lianghua He <helianghua@tongji.edu.cn>.

_Proceedings of the 43^{rd} International Conference on Machine Learning_, Seoul, South Korea. PMLR 306, 2026. Copyright 2026 by the author(s).

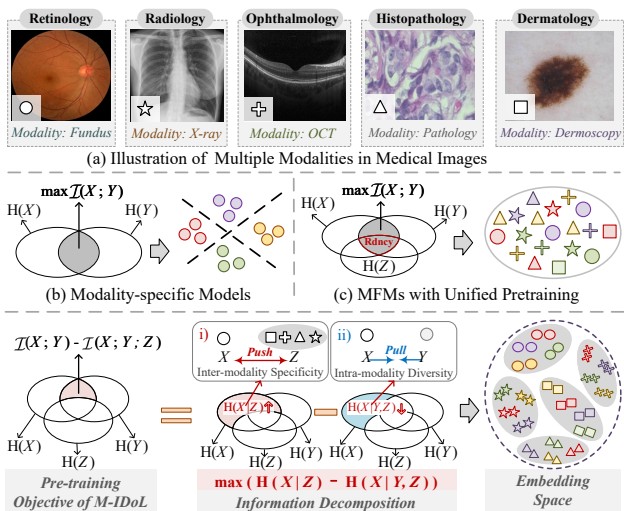

_Figure 1._ (a) Medical images exhibit inter-modality specificity and intra-modality diversity. (b) Modality-specific models excel in intra-modality diversity by focusing on stable imaging statistics. (c) MFMs suffer from information ambiguity due to uniform maximization of redundancy (Rdncy.) information across modalities. (d) M-IDoL mitigates ambiguity via information decomposition, enhancing modality-specific and diverse representations. $X$ and $Y$ denote representations of two augmented views, $Z$ denotes representations from modalities distinct from $X$ and $Y$. $\mathcal{I}(\cdot;\cdot)$ is mutual information, $H(\cdot)$ is information entropy. Shapes (e.g., star, circle) indicate modalities; colors indicate intra-modality semantics (e.g., diseases). t-SNE clusters for (b)–(d) are in Fig. 5.

et al., 2024; M. H. Nguyen et al., 2023; Liu et al., 2025b). A prevailing strategy for current MFMs is unified Contrastive Learning (CL) (Caron et al., 2021), where a single model is jointly optimized across medical imaging modalities (e.g., fundus, X-ray and dermoscopy) by maximizing Mutual Information (MI) between a pair of augmented images (Yan et al., 2025; Khattak et al., 2024). Built on large-scale CL pre-training, MFMs are encouraged to learn universal representations that can be effectively adapted to a wide range of downstream clinical tasks, such as classification and segmentation (M. H. Nguyen et al., 2023).

Despite their empirical scalability, the effectiveness of these unified MFMs is constrained due to the neglect of visual discrimination across medical imaging modalities (Chopra

et al., 2025; Ye et al., 2024). Specifically, as shown in Fig. 1(a), medical images are inherently heterogeneous, with each modality providing distinct anatomical information (M. H. Nguyen et al., 2023). For example, dermoscopy distinguishes early melanoma from benign nevi by subtle pigment irregularities, whereas lung-nodule radiographs rely on faint opacities and structural distortion. This fine-grained visual discrimination emphasizes the importance of *inter-modality specificity* and *intra-modality diversity* in medical multimodal representations (Chopra et al., 2025). However, prevailing MFMs are largely modality-agnostic, implicitly forcing a single embedding space to accommodate all modalities under a uniform CL objective (Khattak et al., 2024; M. H. Nguyen et al., 2023; Xie et al., 2024).

This limitation can be termed **information ambiguity**. To be specific, in modality-specific CL pre-training, as shown in Fig. 1(b), MI maximization primarily reduces prediction uncertainty caused by augmentation noise. This encourages the model to focus on stable statistical patterns that support fine-grained semantic discrimination, such as small lesions, subtle texture changes, and boundary variations, thereby allowing the representation to achieve intra-modality diversity (Zhou et al., 2023a; Liu et al., 2025a; Taher et al., 2024; Yan et al., 2025; Vorontsov et al., 2024). However, in unified pre-training, current MFMs typically apply the same CL objective to multi-modality data (M. H. Nguyen et al., 2023; Khattak et al., 2024). As shown in Fig. 1(c), this causes MI maximization to uniformly reduce predictive uncertainty from both augmentation noise and modality semantic shift. Such uniform reduction inevitably maximizes inter-modality mutual information, which is widely regarded as modality-shared redundancy in multimodal learning ('Rdncy.' in Fig. 1(c)) (Xin et al., 2025; Wollstadt et al., 2023; Bell, 2003). As a result, MFMs bias their training objective toward learning semantically shared representations across modalities. This bias finally undermines fine-grained semantic discrimination, resulting in information ambiguity that degrades both inter-modality specificity and intra-modality diversity.

A straightforward way to alleviate this information ambiguity is to remove modality-shared redundancy from the CL objective, i.e., to maximize intra-modality MI without interfering with the representation learning of other modalities (Fig. 1(d), left). However, in practice, this requires estimating a *trivariate* MI term ($\mathcal{I}(X;Y;Z)$ in Fig. 1(d)) among the representations of a pair of augmented images ($X$ and $Y$), and representations from other modalities ($Z$). Such multivariate MI estimation is difficult and unstable at MFM scale, especially in high-dimensional embedding spaces (Tschannen et al., 2019; Hjelm et al., 2018).

To address this challenge, we propose M-IDoL, a self-supervised _M_edical foundation model that introduces _I_nformation _D_ecomposition to enhance modality specificity

and diversity in multim_o_dal representation _L_earning. Specifically, as shown in Fig. 1(d), M-IDoL decomposes the multivariate MI into two entropy-based objectives that explicitly encourage the model to i) *maximize inter-modality information entropy* (i.e., $H(X|Z)$ in Fig. 1(d)) by increasing the entropy of $X$ given $Z$, ensuring that each representation retains information that is not predictable from other modalities, thereby enhancing modality specificity, and ii) *minimize intra-modality predictive uncertainty* (i.e., $H(X|Y,Z)$ in Fig. 1(d)) by reducing uncertainty between paired views of the same image caused by augmentation noise rather than semantic shifts across modalities, thus improving fine-grained discrimination per modality to enhance representation diversity.

In practice, inspired by the success of Mixture-of-Experts (MoE) in multimodal learning (Chopra et al., 2025; Fang et al., 2025; Xin et al., 2025), M-IDoL optimizes these two objectives by designing a MoE projector. Specifically, to optimize the first objective, M-IDoL introduces a routing-consistency loss to enforce a balanced expert assignment. This pushes multimodal representations into distinct MoE subspaces, promoting visual discrimination and thereby enhancing modality specificity (Fig. 1(d)-i). To optimize the second objective, M-IDoL introduces an intra-modality contrastive loss that pulls same-modality representations within each MoE subspace. This reduces augmentation-caused predictive uncertainty beyond the variance contributed by other-modality representations, promoting modality diversity (Fig. 1(d)-ii). Together, these two losses enable M-IDoL to reduce modality-shared redundancy, thereby mitigating the information ambiguity of unified CL pre-training.

In summary, we make the following contributions:

- We propose M-IDoL, a self-supervised MFM that introduces information decomposition to mitigate information ambiguity by explicitly removing modality-shared redundancy in unified multimodal CL pre-training.

- Via information decomposition, M-IDoL introduces an MoE projector that i) encourages modality-specific representations by maximizing inter-modality information entropy, and ii) increases per-modality diversity by minimizing intra-modality predictive uncertainty.

- Pre-trained on 1.15M unlabeled medical images across five modalities (X-ray, fundus, OCT, dermoscopy, pathology), M-IDoL effectively learns modality-specific and diverse representations, consistently outperforming existing MFMs on 21 downstream datasets.

## 2. Related Work

**Medical Foundation Models.** MFMs have shown impressive performance across medical fields through large-scale self-supervised pre-training. For example, LVM-Med

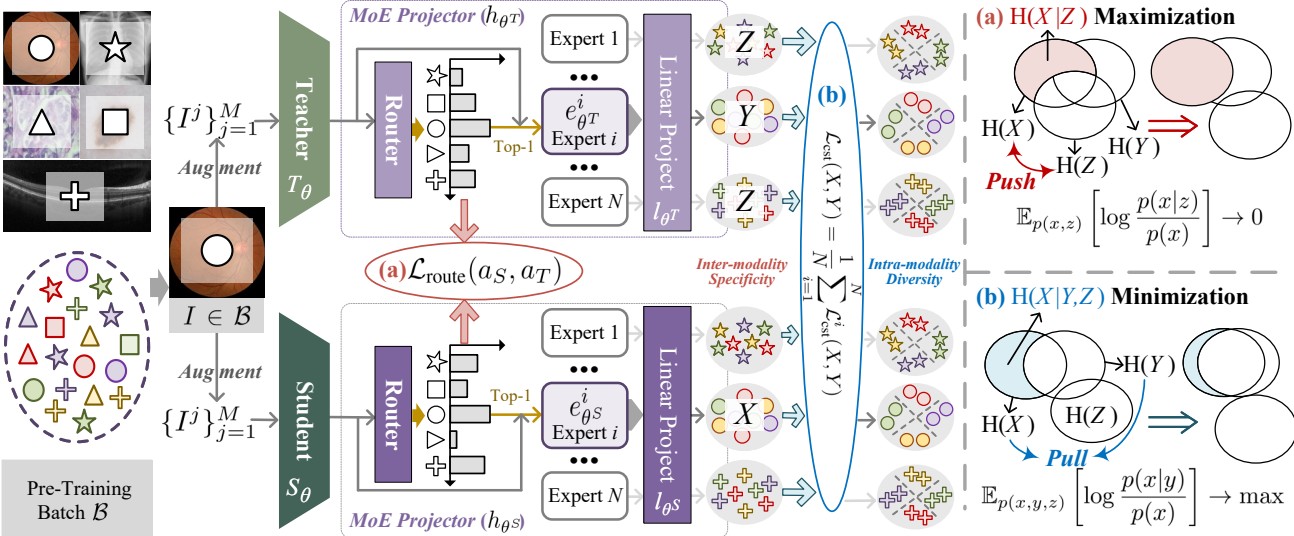

*Figure 2.* Left: Overview of M-IDoL. Via information decomposition, M-IDoL optimizes two objectives: (a) the routing-consistency loss $\mathcal{L}_{\text{route}}$, which learns modality-separable MoE subspaces to maximize $H(X|Z)$ for inter-modality specificity, and (b) the intra-modality contrastive loss $\mathcal{L}_{\text{cst}}$, which promotes fine-grained discrimination within each modality to minimize $H(X|Y,Z)$ for intra-modality diversity. Right: (a) Inter-modality Entropy Maximization ($H(X|Z)$) and (b) Intra-modality Uncertainty Minimization ($H(X|Y,Z)$).

(M. H. Nguyen et al., 2023) learns universal representations from over 1M medical images across 55 datasets, UniMed (Khattak et al., 2024) is trained on 5.3M image–text pairs, and Unimiss+ (Xie et al., 2024) enhances cross-domain representation by augmenting unpaired images. However, these MFMs typically suffer from information ambiguity due to insufficient modeling of medical imaging heterogeneity (Chopra et al., 2025). Recent advances like MedCoss (Ye et al., 2024) and CoSMIC (Liu et al., 2025b) use continual learning to obtain modality-specific features. While effective, they require multi-stage pre-training, which is computationally expensive. In contrast, the proposed M-IDoL introduces a novel information decomposition strategy to remove modality-shared redundancy, efficiently mitigating information ambiguity in a unified pre-training paradigm.

**Mixture-of-Experts.** MoE architectures have shown strong scalability by leveraging sparse conditional routing (Lepikhin et al., 2020; Chopra et al., 2025). More recently, they have been adopted in multimodal learning to enable effective information fusion across heterogeneous modalities (Xin et al., 2025; Fang et al., 2025). In the medical domain, MoE designs largely focus on text-driven semantic alignment (Chopra et al., 2025; Zhu et al., 2024; Rubel et al., 2025) and downstream multimodal prediction (Peltekian et al., 2025; Huy et al., 2025; Chen et al., 2025). FuseMoE (Han et al., 2024) introduces MoE for fleximodal fusion and demonstrates that expert-based routing can effectively accommodate missing or varying modality combinations. Similarly, Flex-MoE (Yun et al., 2024) models arbitrary modality combinations through a flexible MoE design, improving robustness under diverse multimodal input settings.

Moreover, M$^4$oE (Wu et al., 2025) proposes a multimodal multitask MoE framework for medical diagnosis, which dynamically learns modality-specific and shared information to enable adaptive fusion. Despite these advances, existing MoE-based multimodal methods mainly emphasize cross-modal fusion for downstream prediction, overlooking the potential of modality-adaptive specialization in visual representation learning. We innovatively propose an MoE projector that enables the visual encoder to adaptively capture modality specificity and diversity without modality priors, improving visual representation learning in multimodal self-supervised pre-training.

## 3. Method

To mitigate information ambiguity in MFMs, we propose M-IDoL, a self-supervised MFM that enhances medical visual representations with modality specificity and diversity via information decomposition. The overall architecture of M-IDoL is shown in Fig. 2. In the following, we first present the formulation of information decomposition (Sec. 3.2), and then detail how M-IDoL implements this mechanism with an MoE projector to enhance inter-modality specificity and intra-modality diversity during pre-training (Sec. 3.3).

### 3.1. Problem Setup

M-IDoL performs self-supervised CL pre-training with a siamese visual encoder (student $S_\theta$ and teacher $T_\theta$). Both visual encoders are equipped with an $N$-expert MoE projector ($h_{\theta S}$ and $h_{\theta T}$), where each expert is trained to specialize in a single imaging modality. As shown in Fig. 2 left, given an

unlabeled image $I \in \mathcal{B}$, M-IDoL generates $M$ augmented views $\{I^j\}_{j=1}^{M}$ and obtains representations $X = \{x_i^j\}_{j=1}^{M}$ and $Y = \{y_i^j\}_{j=1}^{M}$, where $x_i^j = h_{\theta S}^i (S_\theta(I^j))$ and $y_i^j = h_{\theta T}^i (T_\theta(I^j))$. Here, $\mathcal{B}$ is a minibatch that contains shuffled images from different modalities. $h_{\theta S}^i$ and $h_{\theta T}^i$ are defined as the $i$-th expert selected by the student and teacher MoE projectors for $I^j$, respectively. We assume $\{z_n\}_{n=1}^{N} \setminus \{z_i\}$ as the set of representations from modalities distinct from $X$. Here, $z_n = h_{\theta T}^n (T_\theta(I'))$, $I' \in \mathcal{B}$ denotes an image in the batch with $I' \neq I$. The pre-training objective of M-IDoL is to maximize intra-modality mutual information between $X$ and $Y$ while avoiding interference from representations ($Z$) of other modalities:

$$\max \quad \mathcal{I}(X;Y) - \mathcal{I}(X;Y;Z). \tag{1}$$

### 3.2. Information Decomposition

Traditional MFMs often suffer from *information ambiguity* by uniformly maximizing redundant information across heterogeneous medical imaging modalities (e.g., X-ray, fundus, and pathology) (Khattak et al., 2024; Chopra et al., 2025). This uniformity blends multimodal representations into a single embedding space, degrading inter-modality specificity and intra-modality diversity that are essential for fine-grained diagnosis. To mitigate this issue, M-IDoL explicitly removes the redundancy term ($\mathcal{I}(X;Y;Z)$) from the pre-training objective ($\mathcal{I}(X;Y)$), as formulated in Eq. 1. Since directly estimating multivariate MI is difficult (Tschannen et al., 2019; Hjelm et al., 2018), we introduce *Information Decomposition*, which reformulates Eq. 1 into an *entropy-based expression* for model optimization.

Specifically, in information theory (MacKay, 2003; Bell, 2003), the bivariate MI $I(X;Y)$ is defined as:

$$\mathcal{I}(X;Y) = \mathbb{E}_{p(x,y)}\left[\log \frac{p(x,y)}{p(x)p(y)}\right], \tag{2}$$

and trivariate MI $\mathcal{I}(X;Y;Z)$ (Bell, 2003) is:

$$\mathcal{I}(X;Y;Z) = \mathbb{E}_{p(x,y,z)}\left[\log \frac{p(x,y)\,p(x,z)\,p(y,z)}{p(x,y,z)\,p(x)\,p(y)\,p(z)}\right]. \tag{3}$$

To facilitate estimation of Eq. (3), we first express $\mathcal{I}(X;Y;Z)$ in terms of bivariate MI. Concretely, by multiplying both the numerator and denominator by $p(x)$, the log-ratio in Eq. (3) can be rewritten as

$$\log\left[\frac{p(x,y)}{p(x)p(y)} \cdot \frac{p(x,z)}{p(x)p(z)} \cdot \frac{p(x)p(y,z)}{p(x,y,z)}\right], \tag{4}$$

which yields the following decomposition:

$$\mathcal{I}(X;Y;Z) = \mathcal{I}(X;Y) + \mathcal{I}(X;Z) - \mathcal{I}(X;Y,Z). \tag{5}$$

Then, we express the bivariate MI in entropy form:

$$\mathcal{I}(X;Y) = H(X) - H(X|Y), \tag{6}$$

where $H(X)$ denotes the information entropy of $X$, and $H(X|Y)$ quantifies the predictive uncertainty of $X$ given $Y$ (Shannon, 1948; MacKay, 2003; Bell, 2003). Therefore, substituting Eq. (6) into Eq. (5) yields the entropy form:

$$\mathcal{I}(X;Y;Z) = H(X) - H(X|Y) - H(X|Z) + H(X|Y,Z). \tag{7}$$

Finally, substituting Eqs. (6) and (7) into Eq. (1) makes an explicit entropy-based form of the information decomposition objective:

$$\max \quad \mathcal{I}(X;Y) - \mathcal{I}(X;Y;Z)$$
$$\xrightarrow{\text{Decompose}} \max \quad H(X|Z) - H(X|Y,Z). \tag{8}$$

Here, the first term, $H(X|Z)$ (Fig. 2(a) right), denotes the information entropy of $X$ conditioned on the representations $Z$ from other modalities. A larger $H(X|Z)$ indicates that $Z$ is less informative about $X$, suggesting that $X$ retains richer modality-specific information which is not captured by the other modalities. Maximizing this inter-modality information entropy encourages the model to disperse modality-specific semantics, thereby facilitating representation specificity across modalities. The second term, $H(X|Y,Z)$ (Fig. 2(b) right), quantifies the predictive uncertainty of $X$ given its augmented view $Y$ conditioned on $Z$. A smaller $H(X|Y,Z)$ indicates that $Y$ contributes more semantic-invariant information that effectively reduces uncertainty about $X$ in addition to $Z$. Minimizing this intra-modality predictive uncertainty encourages the model to primarily reduce augmentation noise rather than cross-modality semantic shifts, thus promoting fine-grained discrimination of representation diversity per modality.

### 3.3. M-IDoL Pre-training

Based on Sec.3.2, the pre-training objective of M-IDoL is decomposed into two terms: i) maximize inter-modality entropy $H(X|Z)$ to facilitate modality-specific representations, and ii) minimize intra-modality uncertainty $H(X|Y,Z)$ to promote fine-grained semantic diversity within each modality.

**MoE Projector.** Optimizing $H(X|Z)$ and $H(X|Y,Z)$ requires prior knowledge to identify the correct modality of $X, Y$ and $Z$. However, in practice, images in $\mathcal{B}$ are shuffled across modalities and their modality labels are unavailable. To enable the model to adaptively separate latent modalities, we introduce an MoE Projector after the visual encoder inspired by recent advances in multimodal learning (Xin et al., 2025; Chopra et al., 2025). The proposed MoE Projector consists of a router and $N$ experts, where each expert is designed to correspond to a distinct modality. During pre-training, each expert is encouraged to specialize in a single modality, guiding the model to adaptively assign each unlabeled image to its latent modality subspace. For example,

as shown in Fig. 2 left, the student router produces routing probabilities $a_S^j$ of $I^j$ over all experts as:

$$a_S^j = g_\phi^S(S_\theta(I^j)) \in \mathbb{R}^N, \qquad (9)$$

where $g_\phi^S$ is the router network, implemented as a linear layer followed by a scaling function (e.g. softmax) to obtain normalized routing weights. We adopt top-1 routing and select the expert index with the largest routing probability:

$$i = \arg \max_{i \in \{1,\dots,N\}} a_S^{(j,i)}, \qquad (10)$$

indicating that $I^j$ is more likely assigned to the $i$-th latent modality. We then activate only the selected expert to obtain the representation:

$$x_i^j = h_{\theta^S}^i(S_\theta(I^j)) = l_{\theta^S}\left(e_{\theta^S}^i(S_\theta(I^j))\right), \qquad (11)$$

where $e_{\theta^S}^i$ is the $i$-th expert in the student MoE projector corresponding to the $i$-th latent modality and $l_{\theta^S}$ is a linear projection applied to the expert outputs. $a_T$ and $y_i^j$ in the teacher branch are obtained in the same way.

**Inter-modality Entropy Maximization.** To enhance modality specificity in multimodal representations, M-IDoL is designed to maximize inter-modality entropy $H(X|Z)$. Formally, this conditional entropy is defined as:

$$H(X|Z) = \mathbb{E}_{p(x,z)}\left[\log \frac{1}{p(x|z)}\right]. \qquad (12)$$

Inspired by contrastive density-ratio estimation methods (Tschannen et al., 2019; Ma & Collins, 2018), we rewrite Eq. 12 into a density-ratio form:

$$H(X|Z) = -\mathbb{E}_{p(x,z)}\left[\log \frac{p(x|z)}{p(x)}\right] + H(X), \qquad (13)$$

where $H(X) = -\mathbb{E}_{p(x)}[\log p(x)]$ depends only on the marginal distribution $p(x)$ of $I$. We treat it as approximately constant under normalized representations during optimization. Consequently, maximizing $H(X|Z)$ is equivalent to minimizing the expected log ratio, i.e.,

$$\mathbb{E}_{p(x,z)}\left[\log \frac{p(x|z)}{p(x)}\right] \to 0, \qquad (14)$$

which encourages $X$ to be statistically independent of $Z$ (i.e., $p(x|z) \approx p(x)$) within modality-separated embedding spaces (Fig. 2(a) right).

Therefore, to achieve the inter-modality separation promoted by Eq. 14, we introduce a routing-consistency loss $\mathcal{L}_{\text{route}}$ that facilitates the dispersion of multimodal representations in distinct expert subspaces. Specifically, as shown in Fig. 2(a) left, given a set of augmented views $\{I^j\}_{j=1}^M$,

we define $\mathcal{L}_{\text{route}}$ over their corresponding student and teacher routing probabilities ($a_S$ and $a_T$) as:

$$\mathcal{L}_{\text{route}} = -\frac{1}{M(M-1)}\sum_{j=1}^M \sum_{\substack{\gamma=1 \\ \gamma \neq j}}^M \sum_{i=1}^N a_S^{(j,i)} \log a_T^{(\gamma,i)}, \qquad (15)$$

where $a_S^{(j,i)}$ and $a_T^{(\gamma,i)}$ denote the routing probability assigned to the $i$-th expert for $I^j$ and $I^\gamma$, respectively. To balance the expert assignments, we apply the Sinkhorn–Knopp ($Sin.Kno.$) algorithm (Cuturi, 2013) and $SoftMax$ in $g_\phi^T$ and $g_\phi^S$ to obtain $a_T^j$ and $a_S^j$, respectively. In particular, $Sin.Kno.$ enforces a doubly stochastic routing matrix to balance expert assignments across the batch, thereby preventing all images from routing to a single expert. Consequently, optimizing $\mathcal{L}_{\text{route}}$ encourages all augmented views $\{I^j\}_{j=1}^M$ of the same image to activate the same expert, strengthening the modality-specific representations $X$. Meanwhile, images $I' \in \mathcal{B}$ from other modalities activate different experts, producing representations $Z$ in other subspaces. This process effectively pushes $X$ and $Z$ into separate MoE subspaces, thereby encouraging high-entropy visual discrimination for inter-modality specificity.

**Intra-modality Uncertainty Minimization.** To enhance representation diversity within each modality, M-IDoL simultaneously minimizes intra-modality predictive uncertainty $H(X|Y,Z)$ based on Eq. 14. This objective aims to reduce augmentation noise rather than variance from cross-modality shift. Formally, $H(X|Y,Z)$ is defined as:

$$H(X|Y,Z) = \mathbb{E}_{p(x,y,z)}\left[\log \frac{1}{p(x|y,z)}\right]. \qquad (16)$$

Similar to Eq. 13, we rewrite Eq. 16 into a density-ratio form by introducing the $Z$-conditioned marginal $p(x|z)$:

$$H(X|Y,Z) = -\mathbb{E}_{p(x,y,z)}\left[\log \frac{p(x|y,z)}{p(x|z)}\right] + H(X|Z). \qquad (17)$$

During pre-training, the objective of inter-modality specificity in Eq.14, encouraged by $\mathcal{L}_{\text{route}}$, inherently continues to enforce $X$ and $Z$ to be independent. With this independence assumption, we approximate $p(x|z) \approx p(x)$ and $p(x|y,z) \approx p(x|y)$. Consequently, minimizing $H(X|Y,Z)$ by focusing on the stable imaging statistics in each dispersed MoE subspace is proportional to maximizing the expected log ratio:

$$\mathbb{E}_{p(x,y,z)}\left[\log \frac{p(x|y)}{p(x)}\right] \to \max, \qquad (18)$$

which encourages the augmented representation $Y$ to contribute more semantics-invariant information that reduces the predictive uncertainty of $X$ beyond the representation $Z$ from other modalities (Fig. 2(b) right). In practice, this

*Table 1.* Summary of the datasets used in M-IDoL.

| Pre-train | | Downstream | | | |
|---|---|---|---|---|---|
| Dataset | Images | Dataset | Train | Val | Test |
| Retinology (Fundus) | | Retinology (Fundus) | | | |
| EYEPACS(Gulshan et al., 2016) | 88,702 | APTOS(Karthik et al., 2019) | 2,048 | 514 | 1,100 |
| | | Glaucoma(Kim, 2018) | 861 | 218 | 465 |
| AIROGS(De Vente et al., 2023) | 101,442 | PAPILA[1] | 311 | 79 | 98 |
| | | Retina(Ret) | 336 | 84 | 181 |
| Radiology (X-ray) | | Radiology (X-ray) | | | |
| | | NIH(Wang et al., 2017) | 78,484 | 11,212 | 22,424 |
| MIMIC-CXR(Johnson et al., 2019) | 277,827 | CXP(Irvin et al., 2019) | 219,082 | 5,000 | 234 |
| | | ZhangCXR(Kermany et al., 2018) | 4,983 | 250 | 624 |
| | | RSNA(Stein et al., 2018) | 25,184 | 1,500 | 3,000 |
| | | SIIM-ACR(SII) | 7,969 | 1,898 | 1,904 |
| Ophthalmology (OCT) | | Ophthalmology (OCT) | | | |
| OCT2017(Kermany et al., 2018) | 84,484 | OCTID(Gholami et al., 2020) | 316 | 82 | 174 |
| OCT-8C(Subramanian et al., 2022) | 24,000 | TVHL-DME[2] | 779 | 166 | 168 |
| NEH-OCT(NEH) | 16,813 | TVHL-DR[2] | 355 | 80 | 78 |
| | | OCTDL(Kulyabin et al., 2024) | 1500 | 241 | 323 |
| Histopathology (Pathology) | | Histopathology (Pathology) | | | |
| NCH[3] | 107,180 | Mitosis(Farooq et al., 2024) | 16,119 | 3,454 | 3,456 |
| | | BreakHis-binary(Spanhol et al., 2015) | 1,393 | 299 | 303 |
| PanNuke(Gamper et al., 2019) | 7,901 | BreakHis-Multi(Spanhol et al., 2015) | 1,453 | 312 | 316 |
| | | MHIST(Wei et al., 2021) | 2,427 | 378 | 977 |
| Dermatology (Dermoscopy) | | Dermatology (Dermoscopy) | | | |
| ISIC2020(Rotemberg et al., 2021) | 44,108 | ISIC2016(Gutman et al., 2016) | 750 | 150 | 379 |
| | | ISIC2017(Codella et al., 2018) | 2,000 | 150 | 600 |
| ISIC2024(Kurtansky et al., 2024) | 401,059 | HAM10000(Tschandl et al., 2018) | 10,015 | 193 | 1,512 |
| | | ISIC2018(Codella et al., 2019) | 1,796 | 300 | 500 |

*Table 2.* Impact of different components in M-IDoL.

| MoE | $\mathcal{L}_{\text{route}}$ | $\mathcal{L}_{\text{cst}}$ | RSNA | APTOS | OCTID | Mitosis | HAM10000 |
|---|---|---|---|---|---|---|---|
| ✗ | ✗ | ✗ | 87.42 ± 0.57 | 90.36 ± 0.22 | 96.98 ± 0.47 | 86.72 ± 0.34 | 92.77 ± 0.19 |
| ✓ | ✗ | ✗ | 88.31 ± 0.34 | 87.97 ± 0.52 | 96.77 ± 0.59 | 88.64 ± 0.60 | 90.86 ± 0.28 |
| ✓ | ✓ | ✗ | 91.45 ± 0.16 | 93.46 ± 0.32 | 98.84 ± 0.31 | 91.75 ± 0.27 | 96.66 ± 0.19 |
| ✓ | ✓ | ✓ | 93.23 ± 0.22 | 95.19 ± 0.62 | 99.34 ± 0.35 | 95.62 ± 0.24 | 97.96 ± 0.36 |

*Figure 3.* (a) Visualization of routing assignments for 1,000 images per modality. (b) Impact of expert number on downstream tasks.

objective can be achieved by optimizing a lower-bound approximation of Eq. 18 using the InfoNCE loss as proved by (Ma & Collins, 2018; Tschannen et al., 2019).

Therefore, to optimize the intra-modality invariance promoted by Eq. (18), we propose the intra-modality contrastive loss $\mathcal{L}_{\text{cst}}$ based on InfoNCE to pull $X$ and $Y$ closer within each MoE subspace, enabling the reduction of augmentation-caused uncertainty while preserving modality-specific semantic distinctions. As shown in Fig. 2(b) left, let $\mathcal{B}_i$ denote the subset of images in the current minibatch routed to expert $i$. $\mathcal{L}_{\text{cst}}^i$ in InfoNCE form is defined as:

$$\mathcal{L}_{\text{cst}}^i = -\frac{1}{|\mathcal{B}_i|M} \sum_{b=1}^{|\mathcal{B}_i|} \sum_{j=1}^{M} \log \frac{\sum_{k=1,k\neq j}^{M} s(x_i^{(b,j)}, y_i^{(b,k)})}{\sum_{\beta=1}^{|\mathcal{B}_i|} \sum_{k=1,k\neq j}^{M} s(x_i^{(b,j)}, y_i^{(\beta,k)})},$$
(19)

where $x_i^{(b,j)} \in X$ denotes the representation of the $j$-th augmented view of the $b$-th image routed to the $i$-th expert and $s(\cdot, \cdot) = \exp\left(\text{sim}(\cdot, \cdot)/\tau\right)$ converts the cosine similarity between representations with temperature $\tau = 0.04$.

Aggregating over $N$ MoE subspaces, the batch-level $\mathcal{L}_{\text{cst}}$ is:

$$\mathcal{L}_{\text{cst}}(X, Y) = \frac{1}{N} \sum_{i=1}^{N} \mathcal{L}_{\text{cst}}^i(X, Y).$$
(20)

By optimizing this loss, M-IDoL is encouraged to remove augmentation-caused uncertainty per modality rather than

---

[1]PAPILA (2022), a retinal dataset for glaucomatous diagnosis. URL https://doi.org/10.6084/m9.figshare.14798004.v1.

[2]OCT-TVHL (2022), an OCT dataset for diabetic macular edema (TVHL-DME) and diabetic retinopathy (TVHL-DR) diagnosis. https://github.com/Traslational-Visual-Health-Laboratory/OCT-AND-EYE-FUNDUS-DATASET

[3]NCH (2018), a pathology dataset including NCH-100K and NCH-7K. URL https://doi.org/10.5281/zenodo.1214456.

diminish variations of modality shift. This allows representations to effectively capture discriminative semantic details within a stable modality distribution, such as fine-grained anatomical structures and nuanced tissue-level texture, thereby enhancing intra-modality diversity in MFMs.

Summarily, to enhance modality-specific and diverse representations and thereby mitigate information ambiguity in MFMs, M-IDoL synergistically optimizes $\mathcal{L}_{\text{route}}$ and $\mathcal{L}_{\text{cst}}$:

$$\mathcal{L}_{\text{M-IDoL}} = \mathcal{L}_{\text{route}} + \mathcal{L}_{\text{cst}}.$$
(21)

During early pre-training, $\mathcal{L}_{\text{route}}$ rapidly decreases, potentially establishing reliable subspace assignments for $X, Y$ and $Z$. In later epochs, $\mathcal{L}_{\text{route}}$ tends to stabilize and $\mathcal{L}_{\text{cst}}$ begins to be optimized, indicating that the model focuses on learning diverse representations within relatively stable MoE subspaces. Notably, for downstream tasks, only the teacher network $T_\theta$ is retained as the visual encoder, while **the student network $S_\theta$ and MoE projectors are discarded**.

## 4. Experiments

### 4.1. Implementation Details

M-IDoL follows the standard MFM training paradigm of self-supervised pre-training followed by downstream fine-tuning (Ye et al., 2024; M. H. Nguyen et al., 2023).

**Datasets.** M-IDoL is pre-trained on 1,153,516 unannotated images and evaluated on 21 downstream datasets across $N = 5$ medical imaging modalities: Retinology (Fundus), Radiology (X-ray), Ophthalmology (Optical Coherence Tomography, OCT), Histopathology (Pathology) and Dermatology (Dermoscopy) (Table 1). We enforce a no-overlap policy between pre-training and downstream evaluation. More details are provided in Appendix A.

**Pre-training Setting.** We adopt the DINO (Caron et al., 2021) setting, pre-training Swin-B (Liu et al., 2021) visual

*Table 3.* **Comparison with Modality-specific Models.** *The best performance is* **bolded** *and the second best is* underlined. ‡ *denotes a statistically significant improvement ($p < 0.05$) over the second-best results and n.s. denotes no significant improvement.*

**Retinology (Fundus)**

| Methods | APTOS | | Glaucoma | | PAPILA | | Retina | |
|---|---|---|---|---|---|---|---|---|
| | ACC (%,↑) | AUC (%,↑) | ACC (%,↑) | AUC (%,↑) | ACC (%,↑) | AUC (%,↑) | ACC (%,↑) | AUC (%,↑) |
| RETFound (Zhou et al., 2023b) | 92.17 ± 0.35‡ | 94.36 ± 0.14‡ | 90.18 ± 0.38‡ | 94.36 ± 0.19‡ | 87.25 ± 0.26 | 84.04 ± 0.16 | 84.98 ± 0.13‡ | 85.27 ± 0.53 |
| KeepFIT (Wu et al., 2024) | 92.05 ± 0.63 | 94.26 ± 0.61 | 87.62 ± 1.01 | 91.95 ± 0.41 | 86.64 ± 0.19 | 84.88 ± 0.39‡ | 84.06 ± 0.46 | 87.02 ± 0.31‡ |
| FLAIR (Silva-Rodriguez et al., 2025) | 91.99 ± 0.31 | 93.08 ± 0.84 | 89.05 ± 0.34 | 93.99 ± 0.73 | 87.07 ± 0.59‡ | 83.08 ± 0.42 | 85.31 ± 0.39 | 84.28 ± 0.39 |
| M-IDoL(ours) | **93.43 ± 0.85** | **95.19 ± 0.62** | **90.97 ± 0.58** | **95.05 ± 0.29** | **88.30 ± 0.19** | **86.61 ± 0.34** | **88.4 ± 0.26** | **88.89 ± 0.47** |

**Ophthalmology (OCT)**

| Methods | OCTID | | OCTDL | | TVHL-DME | | TVHL-DR | |
|---|---|---|---|---|---|---|---|---|
| | ACC (%,↑) | AUC (%,↑) | ACC (%,↑) | AUC (%,↑) | ACC (%,↑) | AUC (%,↑) | ACC (%,↑) | AUC (%,↑) |
| RETFound (Zhou et al., 2023b) | 96.39 ± 0.25 | 98.72 ± 0.15 | 96.22 ± 0.34 ‡ | 98.19 ±0.09 | 97.86 ± 0.32‡ | 98.22 ± 0.10 | 86.97 ± 0.13 | 89.05 ± 0.21 |
| MIM-OCT (Pissas et al., 2024) | 95.71 ± 0.33 | 97.85 ± 0.29 | 94.09 ±0.27 | 96.46 ± 0.11 | 92.71 ± 0.15 | 97.77 ± 0.41 | 84.68 ± 0.24 | 87.71 ± 0.64 |
| MIRAGE (Morano et al., 2025) | 96.66 ± 0.14‡ | 99.24 ± 0.09‡ | 95.88 ±0.24 | 98.27 ±0.13‡ | 96.42 ± 0.11 | 98.31 ± 0.16‡ | 88.88 ± 0.20‡ | 89.99 ± 0.34‡ |
| M-IDoL(ours) | **97.13 ± 0.21** | **99.59 ± 0.19** | **98.61 ± 0.20** | **99.34 ± 0.35** | **98.21 ± 0.13** | **99.90 ± 0.05** | **90.38 ± 0.24** | **91.68 ± 0.36** |

**Histopathology (Pathology)**

| Methods | BreakHis-Binary | | BreakHis-8C | | Mitosis | | MHIST | |
|---|---|---|---|---|---|---|---|---|
| | ACC | AUC | ACC | AUC | ACC | AUC | ACC | AUC |
| Phikon (Filiot et al., 2023) | 91.11 ± 0.58 | 95.55 ± 0.49 | 92.33 ± 0.64 | 93.33 ± 0.34 | 80.34 ± 0.32 | 92.24 ± 0.45 | 74.92 ± 0.65 | 83.33 ± 0.74 |
| TransPath (Wang et al., 2022) | 90.87 ± 0.41 | 94.20 ± 0.36 | 91.68 ± 0.41 | 93.42 ± 0.50 | 79.20 ± 0.15 | 90.32 ± 0.13 | 76.64 ± 0.45 | 84.75 ± 0.64 |
| UNI (Chen et al., 2024) | 92.64 ± 0.25‡ | 97.39 ± 0.14‡ | 93.64 ± 0.20‡ | 95.66 ± 0.24 | 85.07 ± 0.46n.s. | 93.34 ± 0.25‡ | **82.09 ± 0.16** | 88.97 ± 0.49 |
| Virchow (Vorontsov et al., 2024) | 92.39 ± 0.31 | 96.48 ± 0.25 | 93.02 ± 0.16 | 96.84 ± 0.31‡ | 84.58 ± 0.27 | 92.97 ± 0.10 | 81.24 ± 0.48 | 89.00 ± 0.28‡ |
| M-IDoL(ours) | **93.00 ± 0.25** | **97.84 ± 0.44** | **94.15 ± 0.31** | **97.48 ± 0.22** | **85.33 ± 0.31** | **95.62 ± 0.24** | 81.78 ± 0.08n.s. | **89.46 ± 0.22** |

**Dermatology (Dermoscopy)**

| Methods | ISIC2016 | | ISIC2017 | | HAM10000 | | ISIC2018 |
|---|---|---|---|---|---|---|---|
| | ACC (%,↑) | AUC (%,↑) | ACC (%,↑) | AUC (%,↑) | ACC (%,↑) | AUC (%,↑) | Dice (%,↑) |
| SwAVDerm (Shen et al., 2024) | 82.25 ± 0.33 | 80.00 ± 0.45 | 83.49 ± 0.35 | 86.59 ± 0.32 | 94.48 ± 0.36 | 95.66 ± 0.42 | 86.61 ± 0.54 |
| Panderm (Yan et al., 2025) | **87.88 ± 0.45** | **86.34 ± 0.25** | 85.87 ± 0.11‡ | **91.85 ± 0.19** | 96.13 ± 0.24‡ | 97.13 ± 0.45‡ | 88.02 ± 0.45‡ |
| M-IDoL(ours) | 86.39 ± 0.39‡ | 84.40 ± 0.21‡ | **86.68 ± 0.23** | 88.71 ± 0.31‡ | **97.10 ± 0.29** | **97.96 ± 0.36** | **88.59 ± 0.57** |

**Radiology (X-ray)**

| Methods | NIH | CXP | ZhangCXR | | RSNA | | SIIM-ACR |
|---|---|---|---|---|---|---|---|
| | AUC (%,↑) | AUC (%,↑) | ACC (%,↑) | AUC (%,↑) | ACC (%,↑) | AUC (%,↑) | Dice (%,↑) |
| DiRA (Haghighi et al., 2022) | 82.79 ± 0.05 | 87.46 ± 0.02 | 94.58 ± 0.26 | 96.74 ± 0.34 | 83.64 ± 0.45 | 90.42 ± 0.58 | 79.78 ± 0.68 |
| Adam (Taher et al., 2024) | 83.23 ± 0.24 | 88.14 ± 0.06 | 93.72 ± 0.11 | 97.71 ± 0.26 | 82.79 ± 0.15 | 89.69 ± 0.77 | 75.68 ± 0.72 |
| PCRLv2 (Zhou et al., 2023a) | 83.95 ± 0.09‡ | 88.24 ± 0.22 | 94.88 ± 0.30 | 97.84 ± 0.24 | 84.88 ± 0.36 | 91.98 ± 0.49‡ | 77.36 ± 0.88 |
| AFiRe (Liu et al., 2025a) | 83.69 ± 0.11 | 88.92 ± 0.14‡ | 95.99 ± 0.59‡ | 98.00 ± 0.18‡ | **86.42 ± 0.24** | 90.54 ± 0.38 | **91.68 ± 1.05** |
| M-IDoL(ours) | **84.27 ± 0.41** | **90.09 ± 0.22** | **96.68 ± 0.24** | **99.69 ± 0.03** | 85.13 ± 0.25‡ | **93.23 ± 0.22** | 91.21 ± 0.84n.s. |

encoders for 100 epochs at $224 \times 224$ image resolution on 2 NVIDIA A6000 GPUs. For each input image, we generate $M = 8$ distinct views using random data augmentations. The batch size during pre-training is 64 per GPU. The proposed MoE projector is based on Tutel (Hwang et al., 2023). More details are provided in Appendix B.

**Downstream Setting.** We apply linear probing on Pathology datasets and full supervised fine-tuning for the other four modalities, following the settings used in their respective modality-specific models. The hyperparameter settings of M-IDoL for each dataset are provided in Appendix C.

### 4.2. Ablation Study

Table 2 presents the ablation results for different components of M-IDoL. Row 1 corresponds to the baseline model, which is jointly pre-trained on five medical imaging modalities using a uniform CL objective. When we add MoE projector only (row 2), the model does not exhibit noticeable performance gains. This is mainly because the MoE routes most modalities to a single expert, leading to unbalanced expert routing (Fig. 3(a)-i). In contrast, including $\mathcal{L}_{\text{route}}$ (row 3) shows a progressive improvement. This in-

dicates the effectiveness of $\mathcal{L}_{\text{route}}$ in separating multimodal representations into distinct MoE subspaces (Fig. 3(a)-ii), which enhances inter-modality specificity in MFMs. Including $\mathcal{L}_{\text{cst}}$ further improves downstream performance (row 4), enabling fine-grained semantic discrimination within each MoE subspace and thus enhancing intra-modality diversity. We also evaluate the number of experts in the MoE projector. As shown in Fig. 3(b), more experts bring limited performance improvement, so for training efficiency we use one expert per modality by default.

### 4.3. Downstream Performance

To evaluate the transferability of M-IDoL, we assess its performance on 21 downstream datasets spanning five medical imaging modalities. All models are fine-tuned with our reimplementations using publicly available code, and results are reported as mean ± standard deviation over three random seeds. We calculate p-values between the top two models using a two-sided t-test to assess statistical significance.

**Compared with Modality-specific Models.** To assess the superior performance of M-IDoL, we compare M-IDoL with SOTA modality-specific Models, each pre-trained at

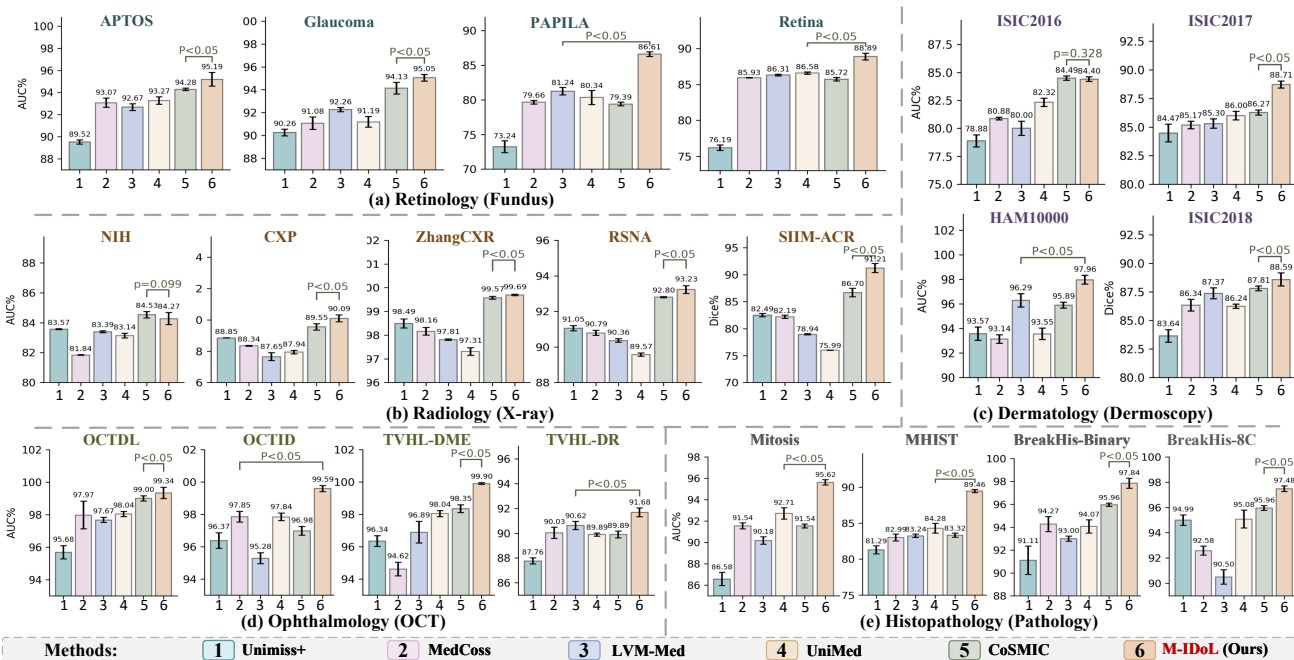

*Figure 4.* Comparison with unified MFMs. Error bars denote standard deviation, the center position reflects the mean performance score (↑, %). $p < 0.05$ indicates that our M-IDoL significantly outperforms the second-best method. Statistical results are in Appendix D.

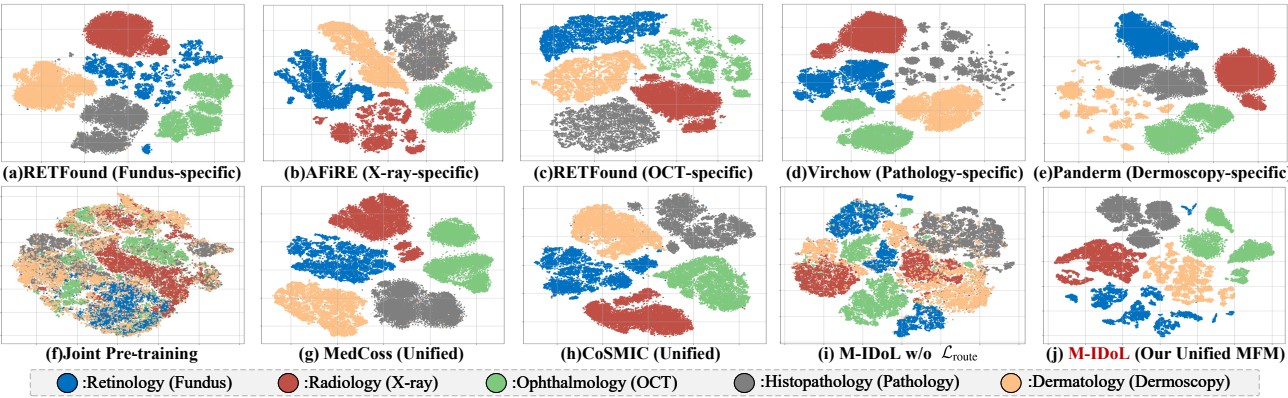

*Figure 5.* t-SNE clusters of representations across five modalities. (a–e) modality-specific models, (f–h) unified MFMs and (i-j) M-IDoL.

scale within its own domain. As shown in Table 3, M-IDoL achieves superior performance on most datasets. Notably, it significantly surpasses RETFound (pretrained on 1.6M fundus images) and MIRAGE (pretrained on 260K OCT images) in Fundus and OCT diagnostic tasks. Although slightly behind Panderm (pretrained on 2.1M dermatology images) on skin disease classification, M-IDoL still shows competitive performance close to the modality-specific state of the art. These results demonstrate that M-IDoL effectively enhances fine-grained semantic discrimination within each modality at a level comparable to modality-specific models, thus promoting intra-modality diversity in unified MFMs.

**Compared with Unified MFMs.** To further evaluate the generalization capability of M-IDoL, we compare it with five existing unified MFMs. As shown in Fig. 4, M-IDoL

achieves significant performance improvements ($P < 0.05$) on most datasets across modalities, and attains competitive performance on NIH ($P = 0.099$) and ISIC2016 ($P = 0.328$) compared to CoSMIC. Particularly, M-IDoL achieves an average improvement of over 6% than Unimiss+ (Xie et al., 2024), LVM-Med (M. H. Nguyen et al., 2023), and UniMed (Khattak et al., 2024), highlighting its superior generalization over existing unified pre-trained MFMs. Besides, compared with MedCOSS (Ye et al., 2024) and CoSMIC (Liu et al., 2025b), which enhance modality specificity in MFMs through continual learning, M-IDoL still delivers over 4% higher performance in OCT and pathology image analysis tasks. These results demonstrate the effectiveness of M-IDoL in achieving stronger visual discrimination across heterogeneous modalities while preserving robust represen-

tational generalization.

**Visualization of Representation Distribution.** We visualize the t-SNE clusters of representations across five modalities in Fig. 5, where 10K images are randomly selected from each modality (following the setting in MedCoss (Ye et al., 2024)). As shown in Fig. 5(a-e), modality-specific models pretrained on a single imaging modality consistently exhibit fine-grained feature discrimination within their target domains, capturing rich intra-modality representational diversity. For instance, in the fundus-specific model RET-Found (Fig. 5(a)), the representations of fundus images expand into multiple well-separated subclusters (blue). In contrast, joint pre-training (Fig. 5(f)) with a uniform CL objective fails to produce clear modality-wise boundaries. This demonstrates significant information ambiguity, where multimodal representations are blended into a shared embedding space, thus leading to suboptimal performance compared with modality-specific models in most diagnostic tasks. Although MedCoss (Fig. 5(g)) and CoSMIC (Fig. 5(h)) are explicitly designed to encourage modality specificity and can separate modalities to some extent, they still exhibit limited fine-grained representational diversity within each modality compared with the modality-specific baselines in Fig. 5(a–e). When M-IDoL is ablated by removing $\mathcal{L}_{\text{route}}$ (Fig. 5(i)), substantial overlap across modalities remains, such as the evident intermixing among the red, orange, and green clusters in the central region. By contrast, M-IDoL with both $\mathcal{L}_{\text{route}}$ and $\mathcal{L}_{\text{cst}}$ (Fig. 5(j)) achieves sharper inter-modality cluster separation and more refined intra-modality feature structuring, providing strong evidence for effective modality-specific and diverse representation learning in joint multimodal medical visual pre-training.

## 5. Conclusion

We propose M-IDoL, a self-supervised MFM that learns modality-specific and diverse representations for robust downstream clinical tasks. M-IDoL innovatively performs information decomposition with an MoE projector to mitigate the information ambiguity by i) maximizing inter-modality entropy and ii) minimizing intra-modality uncertainty. Extensive experiments show that M-IDoL consistently outperforms SOTA MFMs across 21 downstream datasets, highlighting its effectiveness and strong generalization for multi-domain medical image analysis.

## Acknowledgements

Supported in part by the National Natural Science Foundation under Grant 625B2130, in part by the Yeqisun Joint Funds of the National Natural Science Foundation of China under Grant U2441252, in part by the National Natural Science Foundation of China under Grant 62273150, in part by the Changjiang Scholars Program of China, in part by the Computational Biology Program (25JS2840100) of Science and Technology Commission of Shanghai Municipality (STCSM), in part by the Tongji University Medicine-X Interdisciplinary Research Initiative.

## Impact Statement

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

# A. Datasets

## A.1. Retinology

**EYEPACS** (Gulshan et al., 2016) contains $88,702$ retinal fundus images from the United States. This dataset is used for diabetic retinopathy grading on a five-level scale (0–4), where higher grades indicate increasing disease severity.

**AIROGS** (De Vente et al., 2023) includes $101,442$ retinal images from the Dutch national glaucoma screening programme, covering more than $60,000$ individuals across $500$ screening sites. Each image is labeled for referable glaucoma.

**APTOS** (Karthik et al., 2019) comprises $3,662$ retinal images from rural India, curated by Aravind Eye Hospital. Images in this dataset were captured with diverse fundus cameras under varied illumination and time periods, and were annotated into five classes: no DR, mild, moderate, severe, and proliferative DR.

**Glaucoma** (Kim, 2018) consists of $1,544$ fundus photographs from Kim's Eye Hospital for glaucoma severity classification: normal control, early glaucoma, and advanced glaucoma. Images are pre-processed by resizing to $800$ px and cropping around the optic nerve to $240$ px, enabling standardized evaluation for automated glaucoma detection and grading.

**PAPILA** [1] contains $488$ retinal images collected at HGURS (Murcia, Spain) between 2018–2020. Images are categorized as glaucomatous, non-glaucomatous, or suspect, based on comprehensive clinical evaluation.

**Retina** (Ret) is established by Seoul National University (South Korea) to support automated retinal disease detection. It includes $601$ images spanning four categories: normal, cataract, glaucoma, and retinal disease.

In downstream tasks, we follow the RETFound (Zhou et al., 2023b) data splits for APTOS2019, Glaucoma, PAPILA, and Retina.

## A.2. Radiology

**MIMIC-CXR-JPG** (Johnson et al., 2019) contains over $370,000$ unlabeled CXRs. We use $277,827$ frontal-view images for pre-training, without distinguishing between Posteroanterior (PA) and Anteroposterior projection.

**NIH** (Wang et al., 2017) contains $112,120$ frontal-view CXRs from $30,805$ patients with 14 disease labels: Atelectasis (Atel.), Cardiomegaly (Card.), Consolidation (Cons.), Edema (Edem.), Effusion (Effu.), Emphysema (Emph.), Fibrosis (Fibr.), Hernia (Hern.), Infiltration (Infi.), Mass (Mass.), Nodule (Nodu.), Pleural Thickening (P.E.), Pneumonia (Pneu.), and Pneumothorax (PneuX.). For multi-label classification experiments, we use the official split from (Zhou et al., 2023a) with a $7 : 1 : 2$ train/val/test ratio.

**CXP** (Irvin et al., 2019) contains $224,316$ CXRs from $65,240$ patients. We use the official validation set ($234$ images) as the test set. Following the original paper, we perform multi-label classification for Atelectasis (Atel.), Cardiomegaly (Card.), Edema (Edem.), Consolidation (Cons.) and Pleural Effusion (P.E.).

**ZhangCXR** (Kermany et al., 2018) is designed for binary classification, with $1,349$ normal and $3,883$ pneumonia images for training and validation, and $234$ normal and $390$ pneumonia images for testing.

**RSNA** (Stein et al., 2018) includes normal and pneumonia images; we follow the official split: $25,184$ train, $1,500$ val, and $3,000$ test images.

**SIIM-ACR** (SII) comprises CXRs with manual pneumothorax segmentations. We use the $7 : 1.5 : 1.5$ train/val/test split.

## A.3. Ophthalmology

**OCT2017** (Kermany et al., 2018) comprises $84,484$ OCT images, including four categories: Choroidal Neovascularization (CNV), Diabetic Macular Edema (DME), Drusen, and Normal.

**NEH-OCT** (NEH) includes $16,813$ OCT images including Normal, Drusen, and CNV Cases.

**OCT-8C** (Subramanian et al., 2022) contains $24,000$ high-quality OCT images categorized into eight retinal disease classes: Age-related Macular Degeneration (AMD), CNV, central serous retinopathy (CSR), DME, Macular Hole (MH), Drusen, Diabetic Retinopathy (DR), and Normal.

**OCT-TVHL** [2] consists of $1,113$ macular OCT images acquired between 2015 and 2022, intended for diagnosing DME or DR. 1) For TVHL-DME diagnosis, this dataset includes $167$ OCT images with DME and $946$ OCT images without DME. 2) For TVHL-DR diagnosis, $341$ OCT images show no DR, $119$ correspond to non-proliferative diabetic retinopathy, and $53$ correspond to proliferative diabetic retinopathy. For both tasks, we randomly allocate $15\%$ of the data to the test set.

**OCTID** (Gholami et al., 2020) contains over $572$ high-resolution OCT images categorized into distinct pathological conditions, including Normal, MH, AMD, CSR, and DR. Images were acquired using a raster-scan protocol with a $2$ mm scan length and a resolution of $512 \times 1024$ pixels.

**OCTDL** (Kulyabin et al., 2024) comprises over $2,000$ OCT images annotated by disease group and retinal pathology. The dataset includes AMD ($1,231$), DME ($147$), Epiretinal Membrane ($155$), Normal ($332$), Retinal Artery Occlusion ($22$), Retinal Vein Occlusion ($101$), and Vitreomacular In-

terface Disease (76). We randomly split the dataset into training/validation/test sets at a ratio of $7 : 1.5 : 1.5$, ensuring that tiles are derived from disjoint patient groups across splits.

### A.4. Pathology

**NCH-100K** [3] comprises $100,000$ non-overlapping image patches extracted from hematoxylin and eosin (H&E)-stained histopathology slides of human colorectal cancer (CRC) and normal colorectal tissue. Each patch is assigned to one of nine tissue categories: Adipose (ADI), Background (BACK), Debris (DEB), Lymphocytes (LYM), Mucus (MUC), Smooth muscle (MUS), Normal colon mucosa (NORM), Cancer-associated stroma (STR), and Colorectal adenocarcinoma epithelium (TUM).

**NCH-7K** [3] is a histopathology dataset comprising $7,180$ H&E-stained image patches collected from 50 patients with colorectal adenocarcinoma, with no patient overlap with NCH-100K. All tissue specimens were provided by the NCT tissue bank and follow the same preparation and staining protocol as in NCH-100K.

**PanNuke** (Gamper et al., 2019) is a semi-automatically generated benchmark for nuclei instance diagnosis. It provides large-scale, fine-grained nuclei annotations spanning 19 tissue types and five cell categories. The dataset contains $7,901$ images (each of size $256 \times 256$ pixels) and 189,744 labeled nuclei.

**Mitosis** (Farooq et al., 2024) is used to classify three categories of nuclei: mitotic nuclei ($8,703$ images), hard mitotic nuclei ($8,776$ images), and non-mitotic nuclei ($5,550$ images). We randomly reserved $15\%$ of samples from each class as a test set, ensuring class-proportional representation in the evaluation data.

**BreakHis** (Spanhol et al., 2015) is used for Breast Cancer Histopathological Image Classification. It comprises microscopic images of breast tumor tissue collected from 82 patients under four magnification factors (40, 100, 200, and 400). The dataset supports two classification settings: 1) Binary classification: benign vs. malignant tumors. Benign lesions lack hallmarks of malignancy (e.g., pronounced cellular atypia, frequent mitoses, basement-membrane invasion, or metastasis) and typically remain localized with relatively slow growth. Malignant lesions correspond to cancer and may invade adjacent tissue and metastasize to distant sites. 2) Multi-class classification. Both benign and malignant cases can be further categorized into histological subtypes, reflecting differences in cellular morphology that may carry prognostic and treatment implications. BreakHis contains four benign subtypes, *i.e.*, adenosis (A), fibroadenoma (F), phyllodes tumor (PT), and tubular adenoma (TA)—and four malignant subtypes—ductal carcinoma (DC), lobular carci-

noma (LC), mucinous carcinoma (MC), and papillary carcinoma (PC).

**MHIST** (Wei et al., 2021) is a binary classification benchmark for colorectal polyp histology, comprising hematoxylin-and-eosin (H&E) stained formalin-fixed, paraffin-embedded (FFPE) image patches of fixed size ($224 \times 224$ pixels). Images are labeled as Hyperplastic Polyp (HP) or Sessile Serrated Adenoma (SSA). To address class imbalance, we applied random cropping and horizontal/vertical flipping to augment the SSA class, yielding a balanced 1:1 HP-to-SSA ratio in the training set.

### A.5. Dermatology

The International Skin Imaging Collaboration (ISIC) is an international initiative to improve melanoma diagnosis. The ISIC Archive has hosted annual "Grand Challenge" competitions since 2016 with different datasets. The ISIC challenge datasets are summarized as follows:

**ISIC2016** (Gutman et al., 2016) includes 900 dermoscopic lesion images with gold-standard malignant status as ground truth, along with 379 test images in the same format.

**ISIC2017** (Codella et al., 2018) contains 2,000 training images, 150 validation images, and 600 test images for lesion classification (e.g., melanoma and seborrheic keratosis).

**HAM10000** (Tschandl et al., 2018) consists of dermatoscopic images across seven disease categories. We use 10,015 images for training, 193 images for validation and 1,512 images for testing.

**ISIC2018** (Codella et al., 2019) includes 2,596 images for skin lesion segmentation. We randomly split the data into training/validation/test $= 7 : 1 : 2$.

**ISIC2020** (Rotemberg et al., 2021) provides 44,108 dermoscopic training images of unique benign and malignant lesions from 2,000 patients. Malignant diagnoses are confirmed by histopathology, while benign diagnoses are verified via expert consensus, longitudinal follow-up, or histopathology.

**ISIC2024** (Kurtansky et al., 2024) contains 15 mm × 15 mm field-of-view cropped images centered on individual lesions, extracted from 3D total-body photography (TBP). The dataset is curated by hospitals across nine countries and includes 401,059 JPEG lesion crops.

## B. Pre-training Setting

**Data Augmentation.** Given an image $I$, we generate $M = 8$ views, including 2 global views with a crop scale in the range of [0.4, 1.0], and 6 local views with a crop scale of [0.05, 0.4]. We apply the same random image augmentation techniques as in DINO (Caron et al., 2021), such as Hori-

*Table 4.* Detailed configurations for each downstream task

|  | APTOS | Glaucoma | PAPILA | Retina | OCTDL | OCTID | TVHL-DME |
|---|---|---|---|---|---|---|---|
| lr | 5e-4 | 5e-4 | 1e-3 | 8e-4 | 1e-4 | 1e-4 | 1e-4 |
| epoch | 50 | 50 | 50 | 50 | 50 | 50 | 50 |
| shape | 224 × 224 | 224 × 224 | 224 × 224 | 224 × 224 | 224 × 224 | 224 × 224 | 224 × 224 |
| batch size | 64 | 64 | 64 | 64 | 64 | 64 | 64 |
| task | Multi-class cls. | Multi-class cls. | Multi-class cls. | Multi-class cls. | Multi-class cls. | Multi-class cls. | Multi-class cls. |
|  | TVHL-DR | BreakHis-Binary | BreakHis-8C | Mitosis | MHIST | ISIC2016 | ISIC2017 |
| lr | 1e-4 | 1e-4 | 1e-4 | 1e-4 | 1e-4 | 5e-6 | 5e-6 |
| epoch | 50 | 50 | 50 | 50 | 50 | 50 | 50 |
| shape | 224 × 224 | 224 × 224 | 224 × 224 | 224 × 224 | 224 × 224 | 224 × 224 | 224 × 224 |
| batch size | 64 | 64 | 64 | 64 | 64 | 64 | 64 |
| task | Multi-class cls. | Multi-class cls. | Multi-class cls. | Multi-class cls. | Multi-class cls. | Multi-class cls. | Multi-class cls. |
|  | HAM10000 | ISIC2018 | NIH | CXP | ZhangCXR | RSNA | SIIM-ACR |
| lr | 5e-6 | 2e-5 | 3e-3 | 3e-3 | 5e-4 | 5e-4 | 2e-5 |
| epoch | 50 | 160,000it | 200,000it | 200,000it | 50 | 50 | 160,000it |
| shape | 224 × 224 | 512 × 512 | 224 × 224 | 224 × 224 | 224 × 224 | 224 × 224 | 512 × 512 |
| batch size | 64 | samples-per-gpu=2 | 128 | 128 | 64 | 64 | samples-per-gpu=2 |
| task | Multi-class cls. | Segmentation | Multi-label cls. | Multi-label cls. | Multi-class cls. | Multi-class cls. | Segmentation |

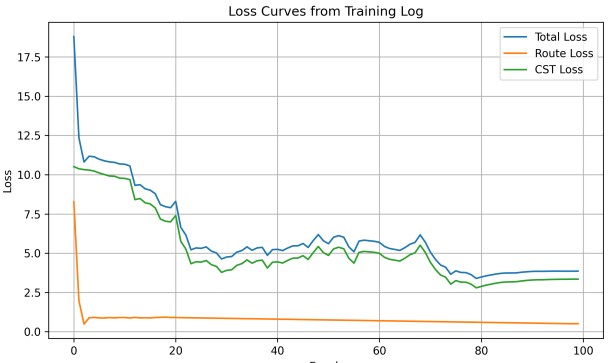

*Figure 6.* Loss Curves from Training Log.

zontal Flip, Color Jitter, Gaussian Blur, and Solarization.

**Architecture.** The M-IDoL pre-training framework consists of two Swin Transformer-Base (Swin-B) visual encoders, i.e., a student encoder $S_\theta$ and a teacher encoder $T_\theta$. Both encoders share the same Swin-B architecture: the input image is split into non-overlapping $4 \times 4$ patches and linearly projected to patch embeddings. The backbone is hierarchical with four stages. Each stage contains a stack of Swin Transformer blocks that alternate window-based multi-head self-attention and shifted-window self-attention, followed by MLP layers with residual connections and normalization. A patch merging layer is inserted between stages to downsample the spatial resolution by a factor of 2 and double the channel dimension. For Swin-B, the stage depths are $[2, 2, 18, 2]$, with embedding dimension 128 and numbers of attention heads $[4, 8, 16, 32]$ (window size $7 \times 7$). The teacher encoder $T_\theta$ is updated as an exponential moving average (EMA) of the student parameters:

$$T_\theta \leftarrow \lambda T_\theta + (1 - \lambda)S_\theta, \qquad (22)$$

where $\lambda \in [0.996, 1)$ is the momentum coefficient, which is scheduled with a cosine annealing strategy during training. During training, gradients are back-propagated only through the student encoder, while the teacher encoder is not updated by back-propagation. During pre-training, the AdamW optimizer ($\beta_1 = 0.9, \beta_2 = 0.95$) is used with an initial learning rate of $1 \times 10^{-4}$ and a weight decay of $0.04$.

**MoE Projector.** The proposed MoE Projector is based on the Tutel Mixture-of-Experts framework (Hwang et al., 2023). The default number of experts in each MoE Projector is 5 with top-1 routing. The router is implemented as a single linear layer that produces routing probability over experts. Each expert is a feed-forward network (FFN). Notably, the MoE projector is used only during pre-training and is removed for downstream evaluation and analysis.

**Optimization of $\mathcal{L}_{\text{route}}$ and $\mathcal{L}_{\text{cst}}$.** Here, we present the loss curves from the pre-training log. As shown in Fig. 6, during early pre-training (first 5 epochs), we found that $\mathcal{L}_{\text{route}}$ rapidly decreases, potentially establishing reliable subspace assignments for modalities $X$, $Y$, and $Z$. This facilitates stable expert specialization by dispersing multimodal representations into distinct MoE subspaces, thereby improving the modeling of inter-modal specificity. In later epochs, $\mathcal{L}_{\text{route}}$ tends to stabilize, where the routing assignments become consistent across augmented views of the same images, while $\mathcal{L}$cst becomes the dominant optimization objective. This shift indicates that the model is gradually guided toward learning diverse and discriminative representations within each subspace.

Overall, the entire training process empirically validates the effectiveness of the proposed information decomposition. By introducing the MoE projector, M-IDoL is able to adaptively disperse representations into multiple special-

*Table 5.* **Comparison with unified pre-training MFMs.** *The best performance is* **bolded** *and the second best is* underlined.

### Retinology (Fundus)

| Methods | APTOS | | Glaucoma | | PAPILA | | Retina | |
|---|---|---|---|---|---|---|---|---|
| | ACC | AUC | ACC | AUC | ACC | AUC | ACC | AUC |
| SwinViT | 89.84 ± 0.57 | 92.30 ± 0.76 | 86.27 ± 0.53 | 91.64 ± 1.41 | 81.37 ± 0.31 | 77.24 ± 0.97 | 77.88 ± 0.14 | 73.17 ± 0.10 |
| DINO | 90.13 ± 0.62 | 92.71 ± 0.29 | 89.54 ± 0.35 | 89.79 ± 0.68 | 81.66 ± 0.14 | 79.81 ± 0.19 | 82.32 ± 0.29 | 76.43 ± 0.29 |
| MAE | 91.21 ± 0.35 | 92.22 ± 0.44 | 88.41 ± 0.72 | 92.14 ± 0.24 | 79.52 ± 0.04 | 73.64 ± 0.07 | 78.31 ± 0.17 | 72.61 ± 0.08 |
| Unimiss+ | 87.52 ± 0.88 | 89.52 ± 0.19 | 86.72 ± 0.37 | 90.26 ± 0.29 | 82.39 ± 0.14 | 73.24 ± 0.85 | 81.32 ± 0.48 | 76.19 ± 0.38 |
| MedCoss | 88.97 ± 0.29 | 93.07 ± 0.42 | 87.24 ± 0.29 | 91.08 ± 0.54 | 77.36 ± 0.34 | 79.66 ± 0.28 | 80.54 ± 0.31 | 85.93 ± 0.04 |
| LVM-Med | 90.31 ± 0.42 | 92.67 ± 0.31 | 88.17 ± 0.54 | 92.26 ± 0.19 | 85.57 ± 0.16 | 81.24 ± 0.53 | 85.27 ± 0.29 | 86.31 ± 0.12 |
| UniMed | 91.34 ± 0.16 | 93.27 ± 0.34 | 88.54 ± 0.64 | 91.19 ± 0.47 | 85.24 ± 0.09 | 80.34 ± 1.01 | 82.88 ± 0.31 | 86.58 ± 0.16 |
| CoSMIC | 92.34 ± 0.04 | 94.28 ± 0.11 | 89.02 ± 0.41 | 94.13 ± 0.52 | 87.37 ± 0.34 | 79.39 ± 0.27 | 86.74 ± 0.58 | 85.72 ± 0.24 |
| M-IDoL(ours) | **93.43 ± 0.85** | **95.19 ± 0.62** | **90.97 ± 0.58** | **95.05 ± 0.29** | **88.3 ± 0.19** | **86.61 ± 0.34** | **88.4 ± 0.26** | **88.89 ± 0.47** |

### Ophthalmology (OCT)

| Methods | OCTID | | OCTDL | | TVHL-DME | | TVHL-DR | |
|---|---|---|---|---|---|---|---|---|
| | ACC | AUC | ACC | AUC | ACC | AUC | ACC | AUC |
| SwinViT | 93.26 ± 0.37 | 97.60 ± 0.07 | 92.37 ± 0.44 | 94.02 ± 0.33 | 89.10 ± 0.76 | 93.15 ± 0.39 | 83.97 ± 0.52 | 88.96 ± 0.11 |
| DINO | 95.04 ± 0.51 | 98.14 ± 0.35 | 94.86 ± 0.62 | 96.35 ± 0.68 | 91.36 ± 0.28 | 92.64 ± 0.64 | 86.25 ± 0.66 | 86.51 ± 0.34 |
| MAE | 93.94 ± 0.31 | 95.98 ± 0.19 | 96.21 ± 0.07 | 95.44 ± 0.39 | 88.64 ± 0.84 | 92.66 ± 0.29 | 81.47 ± 0.25 | 87.98 ± 0.09 |
| Unimiss+ | 93.41 ± 0.46 | 96.37 ± 0.48 | 92.27 ± 0.32 | 95.68 ± 0.41 | 94.24 ± 0.28 | 96.34 ± 0.33 | 84.57 ± 0.27 | 87.76 ± 0.25 |
| MedCoss | 92.88 ± 1.02 | 97.85 ± 0.33 | 94.77 ± 0.31 | 97.97 ± 0.85 | 95.71 ± 0.29 | 94.62 ± 0.42 | 84.75 ± 0.36 | 90.03 ± 0.45 |
| LVM-Med | 91.95 ± 0.62 | 95.28 ± 0.34 | 97.35 ± 0.08 | 97.67 ± 0.16 | 97.62 ± 0.14 | 96.89 ± 0.67 | 85.25 ± 0.64 | 90.62 ± 0.33 |
| UniMed | 94.41 ± 0.36 | 97.84 ± 0.24 | 96.89 ± 0.54 | 98.04 ± 0.16 | 95.66 ± 0.35 | 98.04 ± 0.19 | 87.64 ± 0.29 | 89.89 ± 0.13 |
| CoSMIC | 95.31 ± 0.24 | 96.98 ± 0.27 | 96.21 ± 0.18 | 99.00 ± 0.16 | 96.36 ± 0.33 | 98.35 ± 0.24 | 87.59 ± 0.16 | 89.89 ± 0.27 |
| M-IDoL(ours) | **97.13 ± 0.21** | **99.59 ± 0.19** | **98.61 ± 0.20** | **99.34 ± 0.35** | **98.21 ± 0.13** | **99.90 ± 0.05** | **90.38 ± 0.24** | **91.68 ± 0.36** |

### Histopathology (Pathology)

| Methods | BreakHis-Binary | | BreakHis-8C | | Mitosis | | MHIST | |
|---|---|---|---|---|---|---|---|---|
| | ACC | AUC | ACC | AUC | ACC | AUC | ACC | AUC |
| SwinViT | 89.33 ± 0.43 | 94.38 ± 0.28 | 92.69 ± 0.32 | 95.42 ± 0.54 | 75.07 ± 1.15 | 88.97 ± 0.84 | 75.64 ± 0.52 | 84.09 ± 1.05 |
| DINO | 90.25 ± 0.56 | 96.24 ± 0.11 | 89.89 ± 0.50 | 91.58 ± 0.32 | 78.67 ± 0.27 | 87.88 ± 0.55 | 79.47 ± 1.11 | 86.00 ± 0.59 |
| MAE | 88.27 ± 0.28 | 95.80 ± 0.31 | 91.54 ± 0.45 | 96.66 ± 0.24 | 78.28 ± 0.64 | 89.57 ± 0.41 | 77.89 ± 0.84 | 85.98 ± 0.87 |
| Unimiss+ | 87.69 ± 0.22 | 91.11 ± 1.24 | 89.88 ± 0.98 | 94.99 ± 0.42 | 79.25 ± 0.38 | 86.58 ± 0.62 | 75.99 ± 0.58 | 81.29 ± 0.57 |
| MedCoss | 90.31 ± 0.35 | 94.27 ± 0.66 | 90.18 ± 0.35 | 92.58 ± 0.36 | 80.85 ± 0.34 | 91.54 ± 0.28 | 79.81 ± 0.54 | 82.99 ± 0.48 |
| LVM-Med | 88.00 ± 0.34 | 93.00 ± 0.23 | 90.29 ± 0.87 | 90.50 ± 0.57 | 79.11 ± 0.22 | 90.18 ± 0.34 | 74.21 ± 0.36 | 83.24 ± 0.25 |
| UniMed | 90.54 ± 0.52 | 94.07 ± 0.58 | 91.69 ± 0.39 | 95.08 ± 0.71 | 82.24 ± 0.54 | 92.71 ± 0.54 | 76.19 ± 0.43 | 84.28 ± 0.68 |
| CoSMIC | 91.65 ± 0.25 | 95.96 ± 0.15 | 93.04 ± 0.38 | 95.96 ± 0.18 | 81.64 ± 0.28 | 91.54 ± 0.19 | 77.92 ± 0.34 | 83.32 ± 0.29 |
| M-IDoL(ours) | **93.00 ± 0.25** | **97.84 ± 0.44** | **94.15 ± 0.31** | **97.48 ± 0.22** | **85.33 ± 0.31** | **95.62 ± 0.24** | **81.78 ± 0.08** | **89.46 ± 0.22** |

### Dermatology (Dermoscopy)

| Methods | ISIC2016 | | ISIC2017 | | HAM10000 | | ISIC2018 |
|---|---|---|---|---|---|---|---|
| | ACC | AUC | ACC | AUC | ACC | AUC | Dice |
| SwinViT | 82.80 ± 0.61 | 81.76 ± 0.52 | 81.33 ± 0.22 | 83.70 ± 0.51 | 85.92 ± 0.64 | 93.46 ± 1.10 | 85.29 ± 1.08 |
| DINO | 81.21 ± 0.33 | 82.97 ± 0.43 | 83.69 ± 0.36 | 86.59 ± 0.36 | 84.92 ± 0.28 | 95.68 ± 0.77 | 86.82 ± 0.48 |
| MAE | 82.63 ± 0.54 | 83.19 ± 0.22 | 82.47 ± 0.09 | 83.44 ± 0.21 | 87.24 ± 0.88 | 94.33 ± 0.64 | 86.11 ± 0.24 |
| Unimiss+ | 81.44 ± 0.65 | 78.88 ± 0.54 | 80.00 ± 0.39 | 84.47 ± 0.77 | 92.58 ± 0.59 | 93.57 ± 0.55 | 83.64 ± 0.56 |
| MedCoss | 79.88 ± 0.19 | 80.88 ± 0.12 | 83.11 ± 0.71 | 85.17 ± 0.33 | 90.57 ± 0.55 | 93.14 ± 0.35 | 86.34 ± 0.51 |
| LVM-Med | 83.07 ± 0.25 | 80.00 ± 0.64 | 84.75 ± 0.57 | 85.30 ± 0.41 | 95.67 ± 0.18 | 96.29 ± 0.54 | 87.37 ± 0.48 |
| UniMed | 82.64 ± 0.34 | 82.32 ± 0.36 | 82.11 ± 0.10 | 86.00 ± 0.37 | 94.89 ± 0.81 | 93.55 ± 0.47 | 86.24 ± 0.19 |
| CoSMIC | 82.97 ± 0.24 | **84.49 ± 0.19** | 84.59 ± 0.18 | 86.27 ± 0.22 | 96.09 ± 0.31 | 95.89 ± 0.24 | 87.81 ± 0.21 |
| M-IDoL(ours) | **86.39 ± 0.39** | 84.40 ± 0.21 | **86.68 ± 0.23** | **88.71 ± 0.31** | **97.10 ± 0.29** | **97.96 ± 0.36** | **88.59 ± 0.57** |

### Radiology (X-ray)

| Methods | NIH | CXP | ZhangCXR | | RSNA | | SIIM-ACR |
|---|---|---|---|---|---|---|---|
| | 100% | 100% | ACC | AUC | ACC | AUC | Dice |
| SwinViT | 80.14 ± 0.03 | 86.20 ± 0.04 | 93.69 ± 0.10 | 96.15 ± 0.09 | 82.10 ± 0.12 | 92.00 ± 0.14 | 72.55 ± 0.64 |
| DINO | 80.97 ± 0.01 | 86.78 ± 0.18 | 95.76 ± 0.31 | 96.78 ± 0.04 | 84.21 ± 0.32 | 92.32 ± 0.18 | 70.92 ± 0.76 |
| MAE | 80.96 ± 0.08 | 86.66 ± 0.03 | 93.42 ± 0.26 | 95.88 ± 0.15 | 82.31 ± 0.26 | 91.84 ± 0.14 | 73.58 ± 1.10 |
| Unimiss+ | 83.57 ± 0.03 | 88.85 ± 0.01 | 94.98 ± 0.29 | 98.49 ± 0.19 | 80.88 ± 0.36 | 91.05 ± 0.15 | 82.49 ± 0.31 |
| MedCoss | 81.84 ± 0.02 | 88.34 ± 0.03 | 92.88 ±0.04 | 98.16 ± 0.16 | 80.33 ± 0.13 | 90.79 ± 0.14 | 82.19 ± 0.30 |
| LVM-Med | 83.39 ± 0.07 | 87.65 ± 0.25 | 95.42 ± 0.19 | 97.81 ± 0.03 | 81.19 ± 0.23 | 90.36 ± 0.10 | 78.94 ± 0.12 |
| UniMed | 83.14 ± 0.16 | 87.94 ± 0.11 | 93.57 ± 0.88 | 97.31 ± 0.16 | 82.46 ± 0.24 | 89.57 ± 0.09 | 75.99 ± 0.02 |
| CoSMIC | **84.53 ± 0.22** | 89.55 ± 0.21 | 96.27 ± 0.08 | 99.57 ± 0.06 | 84.19 ± 0.06 | 92.80 ± 0.03 | 86.70 ± 0.79 |
| M-IDoL(ours) | 84.27 ± 0.41 | **90.09 ± 0.22** | **96.68 ± 0.24** | **99.69 ± 0.03** | **85.13 ±0.25** | **93.23 ± 0.22** | **91.21 ± 0.84** |

ized subspaces via $\mathcal{L}_{\text{route}}$, such that representations from different modalities become statistically disentangled across these subspaces. This adaptive decomposition encourages statistical independence between modalities, thereby effectively maximizing the inter-modality information entropy, i.e., $H(X \mid Z)$. Built upon this decomposed representation space, the model further performs representation learning within each independent subspace by pulling together representations of the same image under different augmented views via $\mathcal{L}_{\text{cst}}$. This process reduces prediction uncertainty induced by augmentation noise, without interfering with the learning dynamics of representations in other subspaces, thereby effectively minimizing the intra-modality predictive uncertainty, i.e. $H(X|Y, Z)$. Consequently, the model is encouraged to learn fine-grained semantic discrimination while preserving the independence across modalities. This synergistic optimization demonstrates the potential to enable the model to learn semantically specific and diverse representations, ultimately mitigating information ambiguity in unified MFMs.

## C. Downstream Setting

During downstream fine-tuning, only the pre-trained encoder $T_\theta$ is retained and equipped with a lightweight task-specific head; the student network $S_\theta$ and the two MoE projectors, $h_{\theta S}$ and $h_{\theta T}$, are discarded. Detailed configurations for each downstream task are summarized in Table 4. Below, we describe the specific settings for multi-class classification, multi-label classification, and segmentation.

**Multi-class Classification.** This experiment is implemented using the publicly available codebase of (Zhou et al., 2023b). Images are resampled to a spatial resolution of 256×256. From these resized inputs, random crops are extracted and resized to 224×224. To promote robustness, we apply Mixup and CutMix augmentations throughout training. Optimization is performed using stochastic gradient descent with momentum 0.9 for 50 training epochs on a single NVIDIA A6000 GPU. Models are trained with a batch size of 64 and an initial learning rate is tuned separately for each dataset. The learning rate schedule includes a 10-epoch warm-up phase followed by layer-wise decay, with the learning rate lower-bounded at $1 \times 10^{-6}$.

**Multi-label Classification.** We assess multi-label prediction performance on the NIH and CXP datasets, which require assigning multiple pathology labels to each radiograph. This setting reflects real-world clinical cases where several conditions may be present simultaneously. Optimization is performed using SGD with momentum 0.9, together with a cosine annealing learning rate schedule and a linear warm-up of 50 steps. All experiments use a batch size of 128 and are executed on two NVIDIA A6000 GPUs. We optimize the model using BCEWithLogitsLoss. Input images are

resized to 224×224 with random cropping and horizontal flipping as data augmentation. Validation is carried out every 10 epochs, and the checkpoint with the highest AUC on the validation set is selected for final evaluation.

**Segmentation.** For segmentation tasks, images are resized to 512×512. We fine-tune the model using AdamW optimizer with weight decay 0.05. Learning rates are scaled across the 12 transformer layers with a decay factor of 0.65. Optimization follows a polynomial decay schedule, preceded by a linear warm-up over the first 1,500 iterations, with the learning rate initialized at $2 \times 10^{-5}$ and gradually reduced to zero. Mixed precision (FP16) is performed during training with dynamic loss scaling to stabilize optimization. All segmentation experiments are conducted on four NVIDIA A6000 GPUs for 160,000 iterations. Model checkpoints are written every 10k iterations, while validation is performed at 500-iteration intervals. The checkpoint yielding the highest Dice score on the validation set is selected for final evaluation.

## D. Downstream Performance

Table 5 presents the statistical results of M-IDoL compared with 1) baseline SwinViT (Liu et al., 2021), DINO (Caron et al., 2021) and MAE (He et al., 2022), and 2) unified pre-training MFMs Unimiss+ (Xie et al., 2024), MedCoss (Ye et al., 2024), LVM-Med (M. H. Nguyen et al., 2023), UniMed (Khattak et al., 2024) and CoSMIC (Liu et al., 2025b). The confusion matrix of our proposed M-IDoL is shown in Fig. 7, and the CAM visualizations across five medical imaging modalities are shown in Fig. 8.

## E. Code Availability

The implementation of M-IDoL is based on DINO (Caron et al., 2021) and Tutel (Hwang et al., 2023). We include implementation code in https://github.com/LYH-hh/M-IDoL.

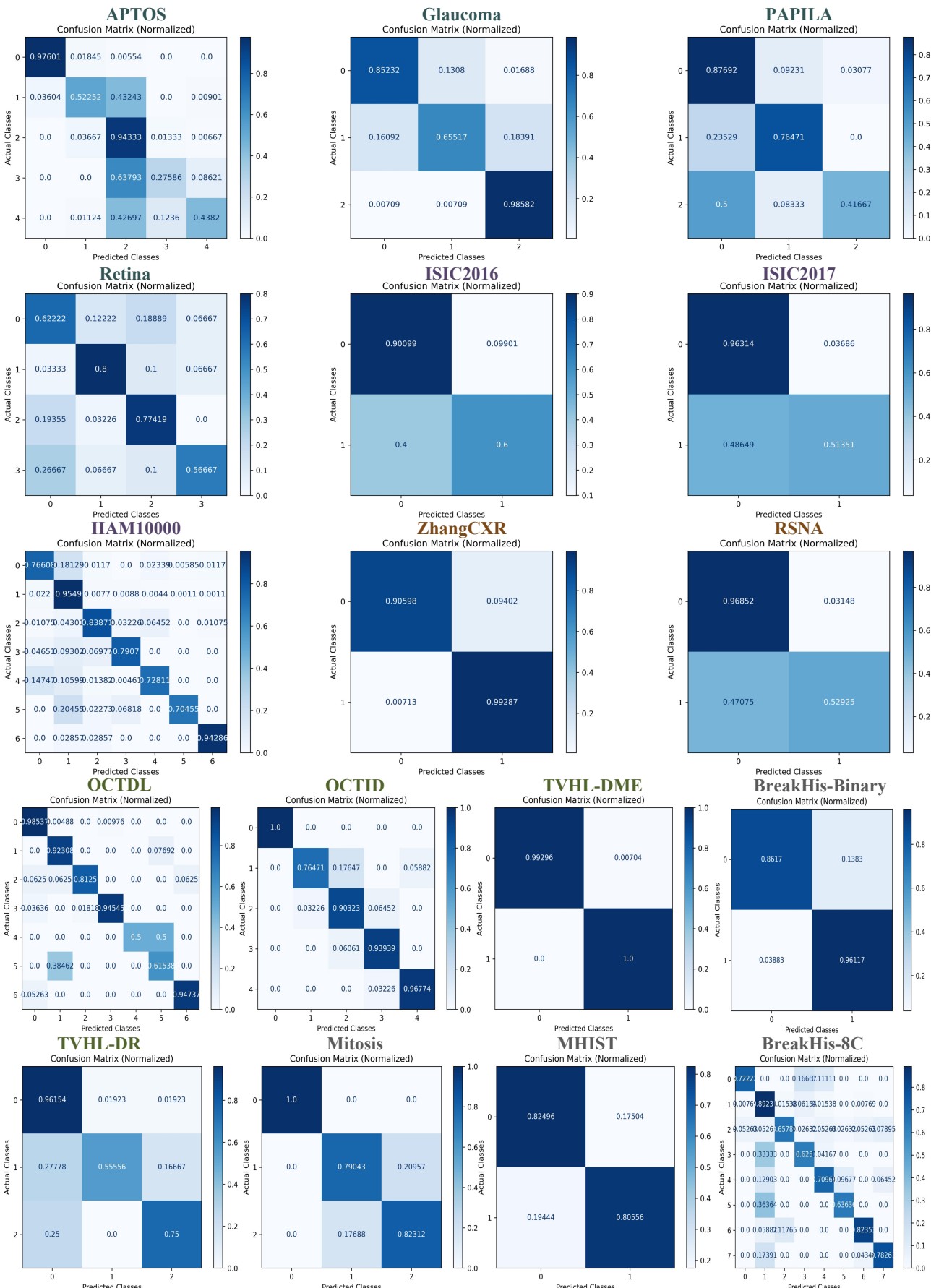

*Figure 7.* Confusion matrices on classification datasets of our proposed M-IDoL.

**(a) Retinology (Fundus)**

**(b) Radiology (X-ray)**

**(c) Dermatology (Dermoscopy)**

**(d) Ophthalmology (OCT)**

**(e) Histopathology (Pathology)**

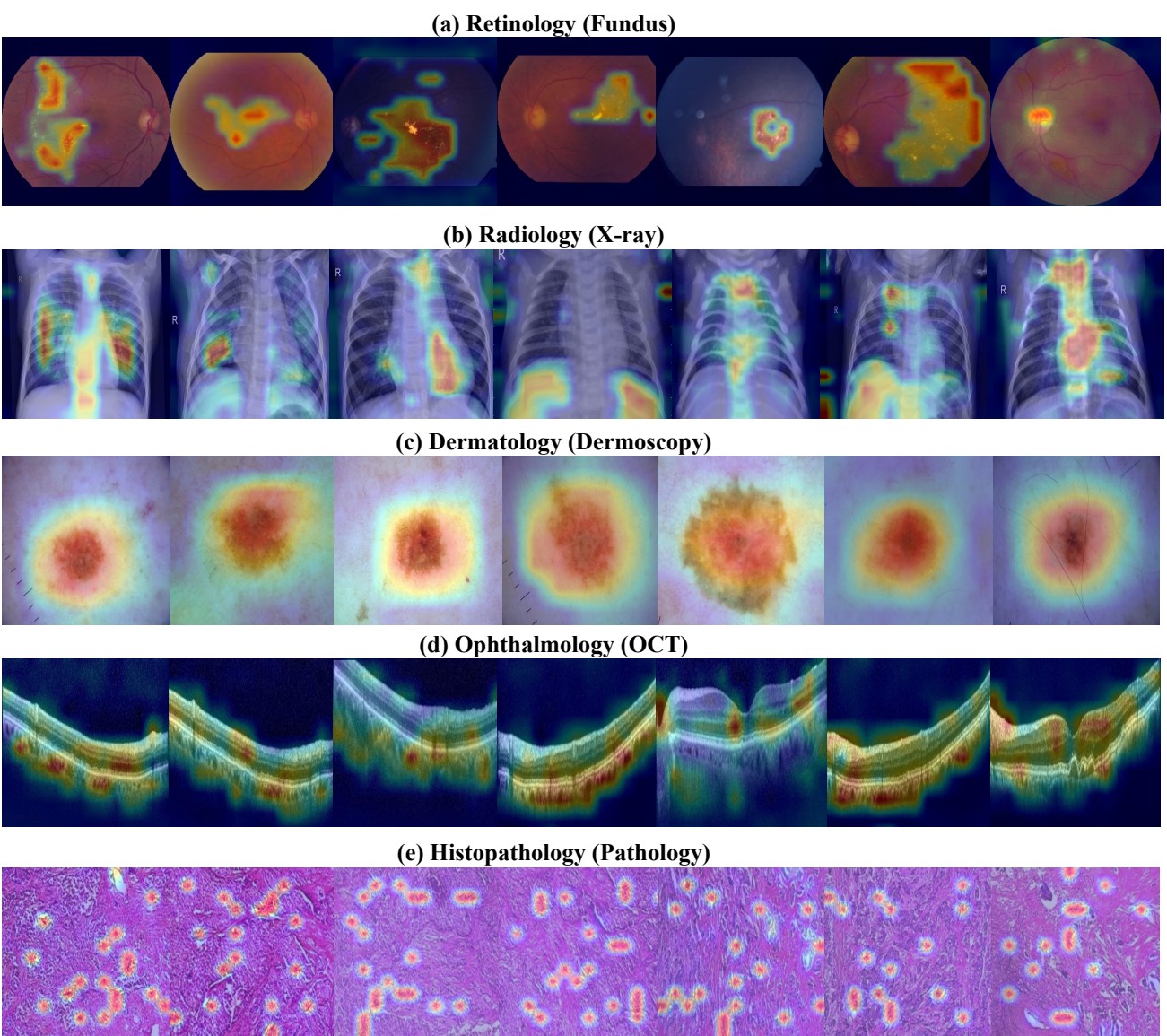

*Figure 8.* Visualization of gradient-weighted class activation mapping (Grad-CAM) across modalities.

