# OpenReview forum: "M-IDoL: Information Decomposition for Modality-Specific and Diverse Representation Learning in Medical Foundation Model"
_ICML.cc/2026/Conference — ICML 2026 regular_

### Official Review · Reviewer_s4bP · 2026-02-21

**Soundness:** 3
**Presentation:** 3
**Significance:** 2
**Originality:** 2
**Overall Recommendation:** 4
**Confidence:** 4

**Summary:**

This paper explores a long-standing challenge in medical multi-modal learning of pre-training a general medical image foundation model with heterogeneous modalities. It starts from the information theories, and aim to address the information ambiguity, through maximizing inter-modality information entropy and minimizing intra-modality predictive uncertainty. To achieve so, they propose a Mixture-of-Experts-based network, integrated with an inter-modality specificity loss and an intra-modality diversity loss. The proposed method's effectiveness have been verified on extensive experiments, across X-ray, fun dus,OCT,dermoscopy, and pathology medical imaging domains.

**Compliance With Llm Reviewing Policy:**

Affirmed.

**Final Justification:**

The rebuttal has addressed most of my concerns, so I will maintain my supportive score.

**Key Questions For Authors:**

See weakness

**Limitations:**

Yes

**Strengths And Weaknesses:**

Strength:

1. This paper is structurally complete, and sells a good story.
2. While medical multi-modal learning + information decomposition has been well studied in supervised learning and multi-modal pre-training, how to adapt them to jointly use heterogeneous imaging modalities to build a general medical foundation model is still under-explored, and this paper has filled this gap.
3. The writing, tables, and visualization are professional.
4. Experiment results are super extensive.



Questions:

Overall, the paper is a good methodology paper and is of very high quality. I guess addressing the following weakness will help further improve the paper and increase the impact of the medical multi-modal learning community.

1. The authors claim that they have adopted the DINO setting, but it's unclear how they combine DINO with the proposed framework. Specifically, DINO has a prototype layer (a bottleneck MLP), is this replaced by the MoE layer? Also, the paper claims to utilize an infoNCE loss to minimize intra-modality specificity, but DINO  uses a self-distillation loss. Have you only adopted the teacher-student dual structure and the ema parts of DINO, and using your own learning objective and network structure? This is a little bit confusing.
2. I understand that the author wants to step-by-step introduce their method, from theoretical insights to specific loss function, and this is how people in medical multi-modal learning do. However, it's still hard for readers to follow. Probably a better practice is to present the key idea in the figure, and highlight motivations and the overall pipeline in the method section.
3. In the related work section, the paper lacks the discussion of works that utilize MoE to address the multi-modal learning problem in medical imaging [1-3], where they also focus on extracting modality-shared information while retaining modality-specific semantics.
4. Some experiments results look weird, and performance gains are sometimes marginal. For example, the AUC and ACC of the ophthalmology downstream tasks are often over 0.97, is the task too easy? Because I note that the performance gain is really marginal, I guess it's better to try whether the proposed method can obtain better performance gain in few-shot learning setting.
5. While the paper is driven by theoretical insights, the information decomposition stuffs are well-studied in recent years, for example [4-5]. It will be better to provide more analysis studies/motivation figures to convince other researchers to trust the claims on the methodology. For example, figure 3 (a) is a good illustration.



[1] Han X, Nguyen H, Harris C, et al. Fusemoe: Mixture-of-experts transformers for fleximodal fusion[J]. Advances in Neural Information Processing Systems, 2024, 37: 67850-67900.

[2] Yun S, Choi I, Peng J, et al. Flex-moe: Modeling arbitrary modality combination via the flexible mixture-of-experts[J]. Advances in Neural Information Processing Systems, 2024, 37: 98782-98805.

[3] Wu C, Shuai Z, Tang Z, et al. Dynamic modeling of patients, modalities and tasks via multi-modal multi-task mixture of experts[C]//The thirteenth international conference on learning representations. 2025.

[4] Zhang Y, Xu Y, Chen J, et al. Prototypical information bottlenecking and disentangling for multimodal cancer survival prediction[J]. arXiv preprint arXiv:2401.01646, 2024.

[5] Liang P P, Cheng Y, Fan X, et al. Quantifying & modeling multimodal interactions: An information decomposition framework[J]. Advances in Neural Inf

---

> ### Author Rebuttal · Authors · 2026-03-29
>
> We are grateful for your effort and constructive suggestions, which greatly help improve our paper. We are encouraged by your positive feedback, which highlights the high quality of our work and the strength of our methodology. By carefully considering your comments, we respond to the questions below.
>
> `Q1: Confusing of DINO setting in M-IDoL`
>
> - **Yes**, as you correctly noted, M-IDoL replaces the prototype layer, i.e. the MLP head, in DINO with our proposed MoE layer.
> - Regarding the loss function, M-IDoL **replaces the self-distillation loss** in DINO with our proposed InfoNCE-based intra-modality contrastive loss.
> - In addition to retaining DINO’s teacher–student dual architecture and EMA mechanism, we also adopt its **centering** operation for the contrastive loss, **data augmentation** strategy, and training settings, such as **weight decay** and the **Adam optimizer**. Other components, including the learning objectives (the MoE projector and the two proposed losses) and the network structure, are designed specifically for our own method.
>
> `Q2: Suggestion on Model Introduction`
>
> We sincerely thank the reviewer for the careful reading and valuable suggestion. We respectfully clarify that the motivation of our method is presented in **Fig. 1**, while the overall pipeline is illustrated in **Fig. 2**. Following your helpful suggestion, we will further emphasize the key idea again in the overall statement of the Method section.
>
> `Q3: Insufficient discussion of related work`
>
> We thank the reviewer for pointing out the relevant references. In response, we expand Sec. 2 to provide a more comprehensive discussion as follows:
>
> - "FuseMoE [1] introduces MoE for fleximodal fusion and demonstrates that expert-based routing can effectively accommodate missing or varying modality combinations. Similarly, Flex-MoE [2] models arbitrary modality combinations through a flexible MoE design, improving robustness under diverse multimodal input settings. Moreover, M$^4$oE [3] proposes a multimodal multitask MoE framework for medical diagnosis, which dynamically learns modality-specific and shared information to enable adaptive fusion. Despite these advances, existing MoE-based multimodal methods **mainly emphasize cross-modal fusion for downstream prediction**."
>
> `Q4: Few-Shot Learning in Ophthalmology`
>
> - As the reviewer correctly observed, the performance in ophthalmology (on OCT images), is indeed consistently high. This is mainly because **OCT images often exhibit more salient visual patterns** than other modalities, making them comparatively easier for models to learn. This observation has also been supported by several previous studies, such as RETFound [Nature'23].
> - Thank you very much for the valuable suggestion to include few-shot experiments in ophthalmology. In response, we compare the performance of M-IDoL against RETFound and CoSMIC in Tab. A, which demonstrates that **M-IDoL still achieves superior performance in the few-shot learning setting**.
>
> Tab. A. Comparison in Ophthalmology under Few-Shot Learning
> ||OCTID||||||OCTDL||||||
> |-|-|-|-|-|-|-|-|-|-|-|-|-|
> ||5-shot||10-shot||16-shot||5-shot||10-shot||16-shot||
> ||ACC|AUC|ACC|AUC|ACC|AUC|ACC|AUC|ACC|AUC|ACC|AUC|
> |RETFound|0.8389|0.8064|0.9198|0.9397|0.9533|0.9802|0.8497|0.7988|0.8919|0.8234|0.8994|0.8889|
> |CoSMIC|0.8452|0.7858|0.8872|0.9572|0.9397|0.9578|0.8499|0.7467|0.8567|0.8011|0.9081|0.8776|
> |M-IDoL|**0.8678**|**0.8289**|**0.9267**|**0.9749**|**0.9583**|**0.9872**|**0.9056**|**0.8118**|**0.9133**|**0.8260**|**0.9128**|**0.9044**|
> ||TVHL-DME||||||TVHL-DR||||||
> ||5-shot||10-shot||16-shot||5-shot||10-shot||16-shot||
> ||ACC|AUC|ACC|AUC|ACC|AUC|ACC|AUC|ACC|AUC|ACC|AUC|
> |RETFound|0.8451|0.9446|0.8778|0.9678|0.9559|0.9721|0.6507|0.7769|0.7231|0.7996|0.7436|0.7977|
> |CoSMIC|0.8219|0.9087|0.8452|0.9324|0.9122|0.9436|0.5907|0.5992|0.6498|0.7035|0.6699|0.7739|
> |M-IDoL|**0.8750**|**0.9548**|**0.9048**|**0.9897**|**0.9583**|**0.994**|**0.6859**|**0.8207**|**0.7628**|**0.8082**|**0.8141**|**0.8240**|
>
> `Q5: Limited analysis of information decomposition.`
>
> - Thank you for pointing out the relevant references [4–5] and for the positive feedback of Fig. 3. Following your suggestion, we will strengthen the analysis around figures to better clarify the motivation of our method.
> - In response, we would like to clarify that, **our work is noval and originality by introducing information decomposition in unified MFM pre-training**. To the best of our knowledge, M-IDoL is the first MFM to explicitly investigate this perspective by decomposing the multimodal mutual-information objective into the entropy-based form H(X|Z)-H(X|Y,Z). This decomposition directly motivates our two objectives for improving inter-modality specificity ($L_{route}$) and intra-modality diversity  ($L_{cst}$). As such, our information decomposition targets solve information ambiguity in the unified pre-training, rather than multimodal interaction modeling or downstream prediction as in prior works [4–5].

---

> > ### Author Rebuttal · Reviewer_s4bP · 2026-04-01
> >
> > All concerns have been addressed

---

> > > ### Author Response · Authors · 2026-04-01
> > >
> > > **Dear Reviewer s4bP,**
> > >
> > > Thank you very much for increasing your confidence score based on your positive evaluation. We are encouraged that you acknowledge our efforts in addressing all your concerns.
> > >
> > > Best regards and have a nice day!
> > >
> > > 4029  Authors

---

### Official Review · Reviewer_JPD2 · 2026-03-09

**Soundness:** 2
**Presentation:** 3
**Significance:** 3
**Originality:** 3
**Overall Recommendation:** 3
**Confidence:** 3

**Summary:**

The manuscript's core contribution concerns addressing the "information ambiguity" problem in unified Medical Foundation Models (MFMs) caused by forcing heterogeneous medical images into a single representation space. To tackle this, the authors proceed to examine the concept of Information Decomposition. They propose M-IDoL, a self-supervised MFM that utilizes a Mixture-of-Experts (MoE) projector to achieve two objectives: inter-modality entropy maximization (routing different modalities to distinct experts) and intra-modality uncertainty minimization (pulling augmented views together within each expert's subspace). Evaluated on 5 modalities and 21 downstream clinical tasks, M-IDoL consistently outperforms existing unified MFMs and achieves highly competitive results against state-of-the-art modality-specific models.

**Compliance With Llm Reviewing Policy:**

Affirmed.

**Final Justification:**

While the paper is promising and the rebuttal adds useful evidence, I remain borderline reject because the key concerns about the gap between the idealized theory and practical routing behavior, as well as the lack of stronger quantitative evidence for modality-level clustering, are only partially resolved

**Key Questions For Authors:**

1. The experiments utilize $M=8$ distinct views for pre-training. Given that $\mathcal{L}_{cst}$ heavily optimizes the consistency among these augmented views, to what extent are the performance gains simply a byproduct of this massive augmentation? How sensitive is the model's downstream performance to the hyperparameter $M$ (e.g., when $M=2$ or $M=4$)?
2. In Table 3, M-IDoL is compared with SOTA modality-specific models like RETFound (pretrained on 1.6M fundus images) and MIRAGE (pretrained on 260K OCT images). Were the baseline unified MFMs and modality-specific models pre-trained from scratch on the exact same 1.15M dataset as M-IDoL?
3. What happens if the number of pre-training modalities is significantly reduced (e.g., down to 2, or even 1)? Does the MoE projector still provide any representational benefits, or does the framework collapse to standard DINO performance?

**Limitations:**

The authors have not adequately discussed the limitations of their approach when applied to paired multimodal datasets (e.g., paired MRI/CT or Fundus/OCT of the same patient), which are extremely common in medical imaging. The proposed method explicitly discards modality-shared information to maximize inter-modality entropy. In a paired-modality scenario, this objective would destructively eliminate critical shared anatomical or pathological semantics. The authors should explicitly discuss this limitation regarding the scope of their framework's applicability.

**Strengths And Weaknesses:**

**Strength**
1. The paper provides a rigorous information-theoretic analysis to formulate the "information ambiguity" problem. By mathematically decomposing the multivariate mutual information, the authors provide a solid theoretical foundation for their proposed loss functions.
2. The adaptation of the MoE architecture as a pre-training projector to achieve both inter-modality entropy maximization and intra-modality uncertainty minimization is clever, effectively avoiding the need for explicit modality labels during pre-training.
3. The experimental design is comprehensive. Ablation studies effectively validate each loss component, and evaluation across 21 diverse downstream tasks demonstrates robust generalization, occasionally surpassing domain-specific SOTA models (e.g., RETFound).

**Weakness**
1. However, the derivation of the intra-modality uncertainty minimization heavily relies on the idealized assumption that $p(x|z) \approx p(x)$ and $p(x|y,z) \approx p(x|y)$. This implies perfect disentanglement of modalities into distinct MoE expert subspaces. Empirical evidence provided in the confusion matrix (Figure 3a) contradicts this idealization, as the routing assignment is far from perfect (e.g., "Dermoscopy" is distributed across three experts). This makes the mathematical simplification weakly supported in practice.
2. The routing consistency loss ($\mathcal{L}_{route}$) only guarantees that augmented views of the *same image instance* are assigned to the same expert. There is no explicit objective ensuring that *different instances from the same modality* are routed to the same expert. The observed modality clustering is merely an emergent property of the domain gap, making the loss design somewhat heuristic despite its strong theoretical motivation.
3. While pitched as a novel MFM framework, the core methodology essentially utilizes an MoE head with Sinkhorn-Knopp balancing to implicitly perform domain-adaptive/multi-task pre-training. Since the MoE projectors are entirely discarded during downstream fine-tuning, the MoE effectively acts as an unsupervised domain clustering head. Similar routing mechanisms are prevalent in multi-task learning, rendering the algorithmic novelty somewhat incremental.

---

> ### Author Rebuttal · Authors · 2026-03-29
>
> We sincerely appreciate your insightful review and valuable suggestions. We especially thank you for pointing out that our work includes rigorous theoretical analysis, clever ablation of the proposed MoE projector and comprehensive experiments.
> We have carefully addressed each of your comments in detail.
>
> **A. Key Questions**
>
> `Q1: Sensitive of M.`
>
> - Thank you for this insightful question.We would like to clarify that enforcing consistency across augmented views is a standard practice in contrastive SSL, not our own trick:). Our $L_{cst}$ is designed to **reduce augmentation uncertainty** within modality-separated MoE subspaces for enhancing intra-modality diversity. Since this objective **requires sufficient augmented samples** for effective contrastive learning, we set **M=8 by following standard settings in existing methods**, such as DINO, rather than using a specially tuned augmentation strategy to boost performance.
> - Following your suggestion, we test sensitivity to the number of views by setting M=2 and M=4. The results are shown in Tab. A, where performance consistently improves as M increases, which is **consistent with the trend reported in DINO**.
>
> **Tab.A Ablation of M**
> |M|APTOS|OCTID|Mitosis|ISIC2017|RNSA|
> |-|-|-|-|-|-|
> |8|95.07|99.52|95.43|88.88|93.13|
> |4|94.27|99.08|93.31|84.71|92.21|
> |2|94.44|98.36|92.17|84.09|91.99|
>
> `Q2: Pretraining Datasets`
>
> No, all compared unified MFMs and modality-specific models use different pre-training datasets based on their original design. We would like to highlight that **this is common in foundation model evaluation**, where methods are typically compared based on their **transferability under the same downstream datasets, rather than being required to share the exact same pre-training setup**. For fairness, we ensure the same downstream datasets, fine-tuning settings, and evaluation protocols for all methods, so the comparison focuses on transfer performance under a consistent setup.
>
> `Q3: Benefit of the MoE projector in single-modality pre-training`
>
> When pre-training is reduced to 1 modality, **the effect depends on the number of experts**. We add this ablation in Tab. B. If we keep 5 experts, the model does not simply collapse. $L_{route}$ still enforces a balanced and view-consistent routing, separating samples into expert subspaces. This facilitates the capture of finer-grained intra-modality representation, resulting an excellent improvement of modality-specific performance (+4% on ISIC206).
> In contrast, if both the number of modality and expert are reduced to 1, the MoE projector degenerates to a single MLP-like head, making the framework collapse to standard DINO performance.
>
> **Tab.B Ablation of different pretraining modalities.**
> ||ISIC2016|ISIC2017|HAM10000|ISIC2018|
> |-|-|-|-|-|
> |5 modalities|84.52|88.88|97.91|88.47|
> |derm-specific+5experts|88.53(+4.01)|93.22(+4.34)|98.39(+0.48)|90.81(+2.34)|
> |derm-specific+1experts|84.22(-0.30)|86.71(-2.17)|94.77(-3.14)|87.66(-0.81)|
>
> **B. Weakness**
>
> `W1: Idealized assumptions in uncertainty minimization`
>
> We would like to clarify that the assumptions of p(x|z)≈p(x) and p(x|y,z)≈p(x|y) are introduced in our theoretical derivation as **idealized objectives to motivate the design** of the loss function, but **not the exact behavior** of the trained model in practice. As the reviewer correctly noted, these equalities would hold exactly **only in the ideal case where the corresponding loss is minimized to 0**. However, **deep models rarely achieve a $L_{loss}=0$**; instead, optimization typically converges once the loss becomes sufficiently small and stable. In our case, the loss decreases from 8.2 to 0.5. Please refer to Sec. B of Appendix.pdf & ./output/log.txt in the Supplementary Material (.zip). Consistent with this, Fig.3a shows that most samples from *Dermoscopy* are routed to their appropriate MoE subspaces, which suggests that "≈" hold approximately, although not perfectly "=".
>
> `W2: Design of routing consistency loss.`
>
> We would like to clarify that our routing loss **does not cause instance-level isolation**, because M-IDoL can still learn a joint routing structure by capturing **shared statistical information across instances** from the same modality. This design effectively encourages same-modality instances to adaptively cluster into the same expert without explicit labels.
>
> `W3: Novelty of  MoE projector`
>
> Thank you for pointing out the novelty of M-IDoL. As you kindly noted, prior work has rarely explored MoE's role in unsupervised pretraining because it relies heavily on label-supervised knowledge. In M-IDoL, we innovatively investigate MoE as a projection head for **label-free multimodal learning** through information decomposition theory.
>
> **C. Limits.**
>
> In paired-modal settings, modality fusion is more critical, which is beyond M-IDoL’s goal of learning specific and diverse cross-modal representations. We will clarify this limitation and explore M-IDoL for paired-modal fusion in future work.

---

> > ### Author Rebuttal · Reviewer_JPD2 · 2026-04-02
> >
> > Thank you for the detailed rebuttal and for providing additional clarifications and experiments.  I still have some concern about
> > - The gap between the idealized independence assumptions and the imperfect routing observed in practice. A more direct quantitative verification would strengthen the claim.
> > - I would also appreciate stronger evidence that the routing mechanism induces modality-level clustering across instances, rather than only view-level consistency for the same sample.

---

> > > ### Author Response · Authors · 2026-04-06
> > >
> > > **Dear Reviewer JPD2**,
> > >
> > > Thank you sincerely for your time and for acknowledging our rebuttal. We would like to provide more clarification on your concerns.
> > >
> > > `1) The gap between independence assumptions and practice routing assignment`
> > >
> > > Thank you for the constructive follow-up. We would like to further clarify that these independence assumptions primarily motivate the design of the route loss, and their practical validity depends on how well the route loss is optimized. More precisely, **the idealized independence assumption holds only when $L_{route}=0$** (as discussed in `W1`). However, during M-IDoL pretraining, **$L_{route}$ decreases only to 0.5 rather than the ideal value of 0**, leading to imperfect routing in practice. This may explain your concern about the gap between idealized and practical routing.
> > >
> > > As you correctly noted,more quantitative evidence would strengthen this claim. In response, we analyze the routing assignments of M-IDoL during pretraining, showing how 1,000 images are gradually routed to the correct experts as the $L_{route}$ is optimized. The overall assignment accuracy, F1, and $L_{route}$ are summarized in **Tab A** and the routing confusion matrices are shown in Fig. a (https://anonymous.4open.science/r/Responce-to-Reviewer-JPD2/). The results show that, at epoch 0, where $L_{route}=8.2$, routing is nearly uniform across experts,indicating no clear modality specialization. By epoch 10, accuracy reaches 75.6%, after which $L_{route}$ stabilizes, showing the gap between independence assumptions and practice routing assignment is minimize.
> > >
> > > **Tab A: The gap between independence assumptions (route loss) and routing accuracy averaged across five experts.**
> > > |Epoch|$L_{route}$|$L_{cst}$|Accuracy|Recall|F1|
> > > |-|-|-|-|-|-|
> > > |0|8.28|10.51|0.2412|0.240832|0.238426|
> > > |1|1.97|10.38|0.6062|0.612965|0.599356|
> > > |3|0.88|10.29|0.6832|0.694644|0.683139|
> > > |5|0.88|10.10|0.7202|0.749135|0.724260|
> > > |7|0.90|9.91|0.6984|0.739942|0.705029|
> > > |10|0.91|9.76|0.7560|0.784607|0.759896|
> > > |20|0.91|7.40|0.7502|0.777305|0.754699|
> > > |45|0.77|4.84|0.7624|0.780550|0.765674|
> > > |60|0.70|4.99|0.7568|0.785399|0.760816|
> > > |85|0.57|3.17|0.7474|0.772090|0.750968|
> > > |99|0.51|3.34|0.7582|0.783098|0.762722|
> > >
> > > To more directly quantify the link between the independence assumptions optimized by $L_{route}$ and practical routing, we compute the correlation between $L_{route}$ and routing accuracy. As shown in Tab.B, the strong negative correlation indicates that **this gap indeed depends on $L_{route}$ optimization**.
> > >
> > > **Tab. B Route Loss vs Overall Accuracy**
> > > |Metric|Value|
> > > |-|-|
> > > |Pearson correlation coefficient|-0.983159|
> > > |Coefficient of determination ($R^2$)|0.966601|
> > > |Spearman rank correlation coefficient|-0.662107|
> > > |Spearman p-value|0.026454|
> > >
> > > While the strong correlation and stable route loss optimization support our claim, **we acknowledge that M-IDoL cannot yet optimize $L_{route}$ to 0 to achieve idealized independence assumptions**. As a result, routing assignment remains mildly imperfect, with accuracy remaining at 75.52 ±0.55. This motivate us a lot to improve the route-loss design in future work.
> > >
> > > `2) Evidence of routing mechanism induces modality-level clustering`
> > >
> > > Thanks for your insightful suggestion. We conduct a t-SNE visualization by ablating M-IDoL with and without  $L_{route}$ or   $L_{cst}$ to demonstrate that the routing mechanism induces modality-level clustering across instances. The visualizations are provided in Fig.b (https://anonymous.4open.science/r/Responce-to-Reviewer-JPD2/).
> > >
> > > In Fig. b(a) (same as Fig. 5(f) in the manuscript), under joint pretraining without the proposed $L_{route}$ and $L_{cst}$, samples from different modalities heavily overlap, and no clear modality-level clustering is observed. In Fig.b(b), when M-IDoL is ablated by removing $L_{route}$, **substantial overlap across modalities remains**, such as the evident *intermixing among the red, orange, and green clusters in the central region* (highlighted by red box in Fig.b).
> > >
> > > By contrast, in Fig. b(c)  (same as Fig. 5(i) in the manuscript) , when $L_{route}$  is introduced in M-IDoL, the embeddings form compact and clearly separated modality-specific clusters, with samples from the same modality gathering in shared regions regardless of instance identity. **This qualitative transition from mixed distributions to globally separable modality-specific clusters** demonstrates that the **routing mechanism learns expert assignments based on modality structure across instances**, rather than only view-level consistency.
> > >
> > > ---
> > > ---
> > > Thank you again for your time, effort, and thoughtful consideration of our work. We greatly appreciate your constructive comments and your positive evaluation, particularly its solid theory and comprehensive experiments. **We would be truly grateful if the explanations above help clarify your concerns**, and we will incorporate your suggestions into the revised manuscript.
> > >
> > > **Best regards and have a nice day!**
> > >
> > > **4029 Authors**

---

### Official Review · Reviewer_on6v · 2026-03-12

**Soundness:** 2
**Presentation:** 2
**Significance:** 3
**Originality:** 2
**Overall Recommendation:** 4
**Confidence:** 3

**Summary:**

This paper proposes M-IDoL, a self-supervised Medical Foundation Model (MFM) that mitigates "information ambiguity" in unified contrastive learning via Information Decomposition. By translating trivariate mutual information into tractable conditional entropy terms, the model uses an MoE projector to maximize inter-modality entropy (separating modalities) and minimize intra-modality uncertainty (enhancing fine-grained features). It demonstrates strong performance across 21 downstream tasks and 5 modalities.

**Compliance With Llm Reviewing Policy:**

Affirmed.

**Final Justification:**

The authors have corrected the statistical claim of their original manuscript and provided additional experiments, and thus I raise my score to Weak Accept but do not champion for it.

**Key Questions For Authors:**

1.	How sensitive is the pre-training stability to the Sinkhorn-Knopp algorithm's hyperparameters? Did you observe mode collapse during the initial epochs before Lroute stabilized?
2.	Regarding the slight underperformance compared to Panderm in the Dermatology task (Table 3), could this be partially attributed to the strict separation of experts? Does this strict separation prevent the dermatology expert from utilizing generalized low-level visual features learned from the massive pool of other modalities?
3.	Could you please correct the absolute claim regarding "P < 0.05 in all datasets" in Section 4.3, given the p-values shown for the NIH and ISIC2016 datasets in Figure 4?

**Limitations:**

The paper would benefit from a clearer discussion of potential limitations, including computational overhead and scalability when the number of experts is large.

**Strengths And Weaknesses:**

Strengths:

1.	Strong Motivation and Practical Value: The paper tackles a critical problem for clinical deployment. By designing a unified framework that rivals or exceeds multiple bespoke models, it effectively overcomes single-modality data bottlenecks while significantly reducing the computational overhead of deploying specialized systems.
2.	Solid Theoretical Foundation: Decomposing the abstract trivariate mutual information into computable conditional entropy terms (H(X|Z) and H(X|Y,Z)) provides a rigorous and sound mathematical basis for the proposed architecture.
3.	Elegant Engineering Implementation: Bridging the theoretical information decomposition objectives with a concrete MoE projector (using routing-consistency and intra-modality contrastive losses) is an innovative and elegant engineering solution.

Weakness：

1.	Overstated Statistical Claims: The claim of achieving "(P < 0.05) in all datasets" (Sec 4.3) is contradicted by Figure 4, where the NIH dataset shows p=0.099 and ISIC2016 shows p=0.328.
2.	Insufficient Mechanistic Analysis: While M-IDoL surprisingly outperforms domain-specific SOTAs (e.g., RETFound), the paper lacks deep analysis on why (e.g., positive transfer of low-level features). Furthermore, it underperforms Panderm in Dermatology. The authors should discuss whether unified architectures inherently struggle against models trained on massive (2.1M), physically unique single-modality data (e.g., color-sensitive dermoscopy).
3.	Scalability and Expert Allocation: The ablation study demonstrates limited performance gains when increasing the number of experts, leading to a default setting of one expert per modality for training efficiency. This implies that the model's capacity does not scale seamlessly with the expert count. If the number of experts must strictly align with the human-defined macro-modalities in the dataset, the "adaptive" nature of the routing mechanism is inherently constrained.
4.	Routing Mechanism Stability: While the routing-consistency loss utilizes the Sinkhorn-Knopp algorithm to prevent expert collapse, there is insufficient analysis regarding the model's sensitivity to Sinkhorn regularization hyperparameters. This is particularly concerning during the early pre-training epochs when feature representations are highly random and unstable.

---

> ### Author Rebuttal · Authors · 2026-03-28
>
> We sincerely thank you for your constructive and insightful feedback. We are grateful for your recognition of the strong motivation, solid theoretical foundation, and elegant implementation of our work. We address your concerns carefully below.
>
> **A. Key Questions**
>
> `Q1: Sensitive of SinkHorn Hyperparams.`
> - We compare different choice in Tab. A. We found that M-IDoL pre-training is **not sensitive to the SinkHorn iteration** (i), using the standard setting of i=3 already suffices (i=30 yields nearly unchanged downstream performance).
> SinkHorn **epsilon (e) is more sensitive** in pretraining. When set e=0.1 (default e=0.05) causes the loss to stall at 14.978965 after 3 epochs, resulting in the  model collapse. Unfortunately, we were unable to test epsilon=0.01 due to the long pre-training time (~4 days on 2×A6000 GPUs). However, prior evidence from SwAV suggests that overly small epsilon may introduce numerical instability. We will further investigate the effect of the epsilon on M-IDoL in future work.
> - **No**, we did not observe mode collapse. ${L}_{route}$ decreased consistently during the initial epochs and then stabilized, without large fluctuations or training stop. The loss logs, training curves, and corresponding analysis (Appendix.pdf Sec B) are provided in the Supplementary Material (.zip); please refer to it for details.
>
> **Tab.A: Ablation of SinkHorn Hyperparams. AUC scores are reported.**
> |Setting|APTOS|OCTID|Mitosis|ISIC2017|RNSA|
> |-|-|-|-|-|-|
> |i=3,e=0.05(default)|95.07|99.52|95.43|88.88|93.13|
> |i=30,e=0.05|94.79|99.49|95.67|89.02|93.18|
> |i=3,e=0.1|72.36|90.85|85.24|57.11|68.92|
>
> `Q2:  Concern about strict separation of experts.`
>
> - The slight underperformance compared to Panderm is more likely due to, as the reviewer kindly pointed out, its much larger dermatology-specific pretraining scale (**2.1M in Panderm  V.S. 0.44M in M-IDoL**), which enables stronger skin-domain priors and finer lesion pattern learning, rather than expert separation in our method.
> - No, we would like to clarify that the setting of one expert per modality **does not enforce strict feature isolation**, because the separation of experts by MoE happens **only in the high-level projection subspace** to reduce cross-modal redundancy, while all modalities still **share the same visual encoder** (lines 191-193 right). Thus, dermatology representations can **still benefit from low-level visual patterns** learned from other modalities.
>
> `Q3:  Concern about P-value statement.`
>
> - Yes, of course. We sincerely thank the reviewer for pointing out this absolute statement in the initial manuscript, and we have revised it to "*M-IDoL achieves significant performance improvements (P < 0.05) on most datasets across modalities, and attains competitive performance on NIH (P = 0.099) and ISIC2016 (P = 0.328) compared to CoSMIC.*"
>
> **B. Weakness**
>
> `W1: Concern about P-value statement.`
>
> We apologize again for this absolute claim, and we sincerely thank the reviewer for pointing it out and helping us improve the paper. We have revised this statement in the manuscript (`Q3`).
>
> `W2: Concern about the performance gap.`
>
> Yes, as discussed in the Introduction (para. 2), unified architectures generally overlook modality heterogeneity in multimodal medical images, and therefore inherently struggle against models trained on single-modality data. To address this issue, M-IDoL proposes information decomposition for multimodal representation learning to enhance modality specificity and intra-modality diversity, thereby **encouraging the model to learn more fine-grained semantic patterns within each modality-specific subspace**, leading to improved performance. This is why M-IDoL can outperform some domain-specific SOTAs (e.g., RETFound). However, M-IDoL indeed shows slight underperformance compared with PanDerm. This is meanly because the **smaller dermatology-specific pretraining scale** in our setting (`Q2`), rather than weaker dermatology-specific specialization of our method.
>
> `W3: Concern about expert adaption`
>
> - We clarify that M-IDoL’s MoE is not a traditional capacity-scaling MoE.
> - The purpose of our MoE is to separate modality-specific subspaces for information decomposition, not to gain performance by simply increasing experts. The limited gains from increasing the number of experts **do not** imply that M-IDoL loses the "adaptive" nature of its routing mechanism; rather, they suggest that once the major modality-level heterogeneity is sufficiently separated, **adding more experts yields diminishing returns.**
>
> `W4: Concern about SinkHorn Hyperparams`
>
> M-IDoL uses 3 iters and 0.05 eps, following default SSL settings in SwAV and AFiRe. We thank the reviewer for this suggestion and have added this ablation in the revised manuscript (`Q1`)
>
> **C. Limitations**
>
> We provide the full model param of M-IDoL as follows, with analysis in Sec4.2. We will add more discussion in the revision.
> |Experts|Para.|
> |-|-|
> |5|174.85M|
> |10|195.85M|
> |20|237.84M|

---

> > ### Author Rebuttal · Reviewer_on6v · 2026-04-03
> >
> > Thank you for the detailed rebuttal. Regarding W2, the explanation still focuses on the data-scale gap, could the authors provide deeper analysis explaining why M-IDoL outperforms domain-specific models like RETFound?

---

> > > ### Author Response · Authors · 2026-04-06
> > >
> > > **Dear Reviewer on6v,**
> > >
> > > Thank you very much for your continued interest and for this constructive question, which inspired us to further investigate the potential of M-IDoL in multimodal medical visual representation learning. We would like to answer your question in detail below.
> > >
> > > We attribute M-IDoL’s superior performance over domain-specific models (e.g., RETFound) to its ability to not only `learn modality-specific representations`, but also `acquire more generalized invariant representations` through unified multimodal pretraining, compared with single-modality models at a comparable overall pretraining data scale like RETFound.
> > >
> > > First, as discussed in our previous rebuttal and as you kindly acknowledged, `M-IDoL is able to effectively learn modality-specific semantics` in the fundus domain by our information decomposition, which helps prevent fundus representations from being degraded by information ambiguity during unified pretraining. This is supported by the t-SNE visualizations in our manuscript, specifically Fig. 5(i) for M-IDoL and Fig. 5(f) for joint pretraining, which show that M-IDoL exhibits clear modality separation.
> > >
> > > Second, on top of preserving modality specificity, `M-IDoL learns better-generalized invariant representations` through multimodal pretraining with a shared visual encoder, which we believe is the **main reason** it outperforms domain-specific models such as RETFound. This advantage of M-IDoL arises from **sharing low-level structural learning across views and modalities through a shared encoder**, while allowing higher-level semantics to remain modality-specific via the MoE projector. As a result, **M-IDoL can learn more generalizable invariant representations than domain-specific models at a comparable overall pretraining scale**, particularly by capturing low-level structural patterns, such as anatomical textures and pathological patterns. While much larger-scale pretraining in domain-specific models (e.g., 2.1M dermoscopic images in PanDerm) may mitigate this limitation, M-IDoL still demonstrates a stronger ability to achieve better performance than RETFound under a comparable pretraining scale (1.15M multimodal images vs. 1.6M fundus images).
> > >
> > > To further support this point, we evaluated the invariancy of representations on paired augmented samples generated from the same fundus image, using CKA as the metric [1]. Specifically, we randomly sampled 1,000 fundus images from the test sets of APTOS, Glaucoma, PAPILA, and Retina, and reported the average layer-wise CKA scores of M-IDoL and RETFound in **Tab. A**. M-IDoL shows a substantially higher average CKA than RETFound (0.7716 vs. 0.3653), particularly in the early and intermediate layers, indicating that **our model learns much more stable cross-view  invariant representations and preserves more shared structural information in low-level features**. Given that early retinal features are largely dominated by local anatomical structures, such as vessels, edges, and lesion boundaries, this result provides further evidence that `M-IDoL captures stronger low-level structural cues in its invariant representations than RETFound.` Therefore, M-IDoL delivers superior performance, outperforming domain-specific models such as RETFound.
> > >
> > > **Tab. A. Layer-wise CKA comparison between M-IDoL and RETFound on 1,000 fundus images using paired augmented views of the same image.**
> > >
> > > |Model Block | M-IDoL CKA | RETFound CKA |
> > > |---|---:|---:|
> > > | 0 | 0.428024 | 0.259788 |
> > > | 1 | 0.455566 | 0.259001 |
> > > | 2 | 0.749242 | 0.268873 |
> > > | 3 | 0.814636 | 0.294237 |
> > > | 4 | 0.756605 | 0.293636 |
> > > | 5 | 0.820220 | 0.294270 |
> > > | 6 | 0.842513 | 0.301249 |
> > > | 7 | 0.861189 | 0.318910 |
> > > | 8 | 0.876053 | 0.330572 |
> > > | 9 | 0.879226 | 0.335246 |
> > > | 10 | 0.882189 | 0.341910 |
> > > | 11 | 0.872677 | 0.353174 |
> > > | 12 | 0.820789 | 0.363276 |
> > > | 13 | 0.800949 | 0.369458 |
> > > | 14 | 0.795988 | 0.376100 |
> > > | 15 | 0.796752 | 0.382097 |
> > > | 16 | 0.798616 | 0.392972 |
> > > | 17 | 0.802244 | 0.404825 |
> > > | 18 | 0.811377 | 0.419445 |
> > > | 19 | 0.822331 | 0.435995 |
> > > | 20 | 0.835529 | 0.456826 |
> > > | 21 | 0.762857 | 0.473503 |
> > > | 22 | 0.748819 | 0.493859 |
> > > | 23 | 0.483649 | 0.546802 |
> > > | **Overall Mean** | **0.771585** | **0.365251** |
> > >
> > > - [1] Similarity of neural network representations revisited, ICML'19
> > >
> > > ---
> > > ---
> > > We sincerely thank you again for your careful review and valuable comments. Your feedback has greatly inspired us to further explore the potential of M-IDoL in unified multimodal pretraining. We are also deeply encouraged by your positive evaluation of our work, particularly regarding its strong motivation, solid theoretical foundation, and elegant implementation. **We have made our best effort to address your concerns**, and we sincerely hope that the responses above help resolve them.
> > >
> > > **Best regards and have a nice day!**
> > >
> > > **4029 Authors**

---

### Official Review · Reviewer_gAZF · 2026-03-13

**Soundness:** 3
**Presentation:** 3
**Significance:** 2
**Originality:** 2
**Overall Recommendation:** 3
**Confidence:** 4

**Summary:**

This paper proposes a self-supervised medical foundation model that enhances multimodal medical representation learning through information decomposition. To address the information ambiguity arising from unified contrastive learning in existing medical foundational models, the authors propose information decomposition to optimize representation learning. After pretraining on 1.15 million medical images (covering five modalities: X-ray, fundus, OCT, dermatoscopy, and pathology), M-IDoL outperforms existing foundational models across 21 downstream clinical tasks.

**Compliance With Llm Reviewing Policy:**

Affirmed.

**Final Justification:**

After considering the paper, the authors’ rebuttal, and the reviewer discussion, I remain at WR. I appreciate the paper’s strong motivation, broad empirical scope across modalities and downstream tasks, and the potentially useful idea of using information decomposition with an MoE projector to encourage modality-specific representations. The rebuttal was helpful in clarifying several implementation details and in adding further analysis, and I understand why it improved some reviewers’ impressions. However, main concerns were only partially resolved. In particular, I am not fully convinced that the reported gains can be cleanly attributed to the proposed decomposition mechanism itself, rather than to broader differences in pretraining setup, data scale, or comparison conditions. I also remain unconvinced that the paper sufficiently validates its central conceptual claim regarding harmful cross-modal ambiguity versus beneficial shared semantics, especially given the limited evidence on tightly controlled comparisons and the still incomplete discussion of settings such as missing modalities at inference. Thus, while I see clear promise and potential significance in the direction, I do not think the current version is yet strong enough in soundness and originality for acceptance, and the paper’s positioning and clarity would also benefit from further refinement. Overall, the rebuttal improved my understanding of the work, but it did not change my evaluation, the key empirical and conceptual claims still need stronger support before the paper can be accepted.

**Key Questions For Authors:**

1. Why does the unified contrastive learning method for existing medical foundation models lead to information ambiguity? Could the authors provide a more detailed theoretical analysis explaining why uniformly maximizing redundant information compromises modality specificity and diversity?
2. When evaluating different modalities, did the authors consider the sample imbalance issue across modality datasets? For example, the significant disparity in the number of fundus images (approximately 88K) and pathology images (approximately 107K) – did this impact the model's generalization ability across different modalities?
3. M-IDoL outperforms 20 foundational models across 21 downstream tasks, but were the pre-training data volumes and computational resources comparable for these models? For instance, UniMed utilized 5.3 million image-text pairs, while M-IDoL employed 1.15 million medical images—does this disparity compromise the fairness of the performance comparison?

**Limitations:**

The authors are encouraged to discuss the performance of M-IDoL on cross-center or cross-device datasets. Medical image data is often affected by variations in acquisition devices and centers, which may impact the model's generalization ability—a critical factor for real-world applications.

**Strengths And Weaknesses:**

Strengths:
1. The paper proposes a novel perspective on information decomposition, effectively addressing the issue of information ambiguity in foundational medical models.
2. Extensive evaluation across five medical imaging modalities and 21 downstream tasks, covering multiple clinical application domains.

Weaknesses:
1. Although the authors identify the issue of information ambiguity, they fail to provide an in-depth analysis of why existing unified contrastive learning methods lead to this problem, leaving the motivation for this work unclear.
2. The mathematical derivation of information decomposition is relatively straightforward.
3. The comparison with some of the latest foundational medical models is insufficiently detailed, particularly in failing to adequately discuss how differences in model scale and pre-training data volume impact performance.
4. The comparison with the SOTA method does not appear to be based on identical training data, which means the results presented in this paper cannot prove that the proposed approach is superior.
5. This work lacks sufficient algorithmic originality, yet this shortcoming is overshadowed by its large-scale pretraining combined with extensive performance evaluation. While it holds significant value, I would prefer to characterize it as an engineering application.

---

> ### Author Rebuttal · Authors · 2026-03-28
>
> Thank you very much for your time and invaluable comments! We appreciate your recognition of M-IDoL’s novelty and the robustness demonstrated through our extensive evaluation. We would like to address your concerns as follows:
>
>  **Key Questions & Weaknesses**
>
> `Q1 & W1: Why unified Contrastive Learning (CL) in MFM causes information ambiguity?`
> - We respectfully clarify that information ambiguity in unified MFMs has already been analyzed in the **Introduction (paras. 2–4)**, and we further illustrate this motivation in **Fig. 1.**
> - In response, we explain that CL aims to reduce predictive uncertainty introduced by data augmentation so as to learn invariant and discriminative representations. However, as explicitly stated in the manuscript (left of lines 59–86), medical images from different modalities are heterogeneous (Fig.1a), whereas the unified CL in existing MFMs **overlook such important modality heterogeneity**(Fig.1c). As a result, the predictive uncertainty is reduction from not only augmentation noise but also meaningful modality-specific semantic variability. This causes **multimodal representations to collapse into a single embedding space**, leading to information ambiguity, as also illustrated by the cross-modality overlap in Fig. 5f.
> - Mathematically, let $X$ and $Y$ denote the representations of two augmented views. CL maximizes the mutual information (MI) $I(X;Y)$ between them (Fig.1b). However, in multimodal pre-training, the model also capture information redundantly shared with another modality $Z$ (Fig.1c). Therefore, max $I(X;Y)$ enforces a three-variable information space as
> $I(X;Y)=\mathbb{E}_{p(x,y,z)}\left[\log \frac{p(x,y\mid z)}{p(x\mid z)p(y\mid z)}+\log
> \frac{p(x,y)\,p(x,z)\,p(y,z)}
> {p(x,y,z)\,p(x)\,p(y)\,p(z)} \right]$. The *first term measures the information between $X$ and $Y$ conditioned on $Z$* and should be maximized to promote intra-modal diversity by removing the shared effect of $Z$. The **second term denotes the three-way MI among $X$, $Y$, and $Z$ (as defined in manuscript Eq.(3) $I(X;Y;Z)$)**, i.e., **the redundant information shared across modalities** that **hurts inter-modal specificity learning**.
> Therefore, unified CL inevitably promotes maximizing $I(X;Y;Z)$, making it unable to distinguish modality-specific invariance from redundancy shared across modalities, thereby compromising both modality specificity and diversity.
>
> We thank the reviewer for highlighting the need for stronger theoretical analysis, and a detailed proof will be added to the revised manuscript.
>
> `Q2: Sample imbalance issue across modality in pre-training.`
>
> - Yes, we have considered the sample imbalance when pre-trianing M-IDoL.
> - However, we found that this imbalance did not degrade M-IDoL’s performance.This is because M-IDoL maps modality-specific representations into separate subspaces, allowing **each modality to focus on its own discriminative patterns despite data imbalance**. Moreover, we found that each modality’s performance was more strongly related to its own data scale than to the data volume of other modalities (Tab. a). Therefore, we collected as much data as possible for each modality, e.g., 190K fundus and 115K pathology images.
>
> **Tab.a: More pretraining data per modality, stronger M-IDoL performance**
> |||Fundus|OCT|Path.|Derm.|X-ray|
> |-|-|-|-|-|-|-|
> |Default|Pretrain data|190144|125297|115081|445170|277827|
> ||Avg. AUC|90.86(+1.19)|96.86(+0.77)|91.83(+0.85)|89.98(+3.49)|91.47(+2.88)|
> |Balenced|Pretrain data|101442|108484|107180|110000|110000|
> ||Avg. AUC|89.67|96.09|90.98|86.49|88.59|
>
> `Q3 & W3 & W4: Concern about pre-training settings in comparison.`
>
> - We would like to clarify that M-IDoL uses different pre-training data, but the same visual encoder scale as most MFMs, such as CoSMIC and MIRAGE, all of which use base-scale Transformer (86–90M param.).
> - We sincerely thanks for your careful review, but it is important to explain that most MFMs (i.e. the models listed in tab.3 and fig.4) are pretrained on different datasets, often involving private data, which makes it **difficult to match pretraining data volumes and computational resources exactly across models**. MFM evaluation mainly relies on **downstream task performance to assess transferability**. As you pointed out, data volume are also important in comparison. In this respect, our model is not disadvantaged in data scale (1.15M samples) or model scale (88M encoder). Moreover, M-IDoL outperforms UniMed (5.3M) with only 1.15M pretraining samples, highlighting its efficiency and robustness.
>
> `W2 & W5: Algorithmic originality`
>
> We would like to clarify that **M-IDoL is original** in exploring the domain gap issue from an information-theoretic perspective. To the best of our knowledge, this is the first MFM that explicitly studies information decomposition for learning modality-specific and diverse representations in unified multimodal pretraining, and its effectiveness is validated by extensive experimental results.

---

> > ### Author Rebuttal · Reviewer_gAZF · 2026-04-04
> >
> > Thank you for the rebuttal.
> > * While you attribute the ambiguity issue to unified CL, could you clarify whether this effect persists under controlled settings where only the loss function is changed while keeping data sampling and augmentations fixed?
> > * The paper assumes redundant cross-modal information is harmful. Could you provide empirical evidence distinguishing harmful redundancy from beneficial shared semantics? How does model perform under missing-modality settings at inference time?

---

> > > ### Author Response · Authors · 2026-04-06
> > >
> > > **Dear Reviewer gAZF**,
> > >
> > > We sincerely thank you for your time in reading our rebuttal. We are pleased that we have addressed most of your concerns. Our detailed responses to the two remaining concerns are below.
> > >
> > > `1)  Does the ambiguity issue persist when only the loss function changes?`
> > >
> > > Thank you for raising this important point. We would like to clarify that, in M-IDoL, the **ambiguity issue can be effectively addressed** by **only** replacing the traditional *unified CL loss* with our proposed *routing-consistency loss* and *intra-modality contrastive loss* , while **keeping all other settings unchanged**, including the data sampling and augmentation strategies.
> > >
> > > **The experiment supporting this point is provided in Fig. 5 of the manuscript.** Specifically, we jointly pretrain a visual encoder with unified CL loss under exactly the same setup as M-IDoL, including the same pretraining data, data augmentation strategy, model architecture, and training hyperparameters, such as weight decay and the Adam optimizer. We then visualize the t-SNE clusters of the encoder jointly pretrained with unified CL in Fig. 5(f) and compare them with those of M-IDoL in Fig. 5(i), where the only difference is the loss function.
> > >
> > > As shown on Fig. 5(f), **joint pre-training with unified CL fails to produce clear modality-wise boundaries**. This demonstrates substantial information ambiguity, where multimodal representations are blended into a shared embedding space.
> > > In contrast, our M-IDoL (Fig. 5(i)) achieves both **sharper inter-modality cluster separation** and **more refined intra-modality feature structuring**. This result demonstrates that the information ambiguity inherent in unified CL can be effectively mitigated in our method by using the proposed losses. Further analysis is provided in Lines 429–438 of the left column and  Lines 421–426 of the right column in the manuscript.
> > >
> > > `2) Evidence on harmful vs. beneficial shared semantics & missing-modality setting`
> > >
> > > Thank you very much for this question. We would like to respectfully explain that there may be a misunderstanding of M-IDoL. Consistent with existing medical foundation models (MFMs) [1-3], **M-IDoL is designed as a pre-training framework to learn modality-specific and diverse representations, rather than the beneficial shared semantics typically emphasized by multimodal fusion models for joint inference.**
> > >
> > > *In this context, we would like to further clarify that our primary focus is **not to distinguish harmful redundancy** from beneficial shared semantics, but to **distinguish multimodal representations** from a blended embedding space by reducing modality-shared redundancy.*
> > >
> > > Specifically, the "cross-modal information" discussed in our paper does **not** refer to the beneficial shared semantics that multimodal fusion models generally aim to preserve. Instead, it refers to **ambiguous modality-mixed** features arising from the over-alignment of heterogeneous modalities. This cross-modal information, as also observed in [1–2], is **redundant** in MFM pretraining, **harms modality-specific** representations, and should therefore be removed.
> > >
> > > *Accordingly, similar to other MFMs [1–3], the missing-modality setting at inference time is **not** directly applicable to M-IDoL, since M-IDoL uses multimodal data **only during pretraining**, whereas downstream adaptation and inference are performed using a single available modality.*
> > >
> > > Following your kindly suggestion, we would like to provide evidence that redundant cross-modal information is harmful for MFM by: i) **the ablation in Tab. 2**, where the unified multimodal baseline, which does not remove modality-shared redundancy, consistently yields the weakest performance (RNSA -5.81, APTOS -4.83, OCTID -2.36, Mitosis -8.90, HAM10000 -5.19). ii) **the routing visualization in Fig. 3(a)**, which shows clearer modality-wise expert separation after redundancy reduction; and iii) **the t-SNE results in Fig. 5**, where unified joint pre-training produces blended shared embeddings, while M-IDoL yields sharper inter-modality separation and finer intra-modality discrimination.
> > >
> > > [1] CoSMIC: Continual Self-supervised Learning for Multi-Domain Medical Imaging via Conditional Mutual Information Maximization (ICCV'25)
> > >
> > > [2] Continual Self-supervised Learning: Towards Universal Multi-modal Medical Data Representation Learning (CVPR'24)
> > >
> > > [3] LVM-Med: Learning Large-Scale Self-Supervised Vision Models for Medical Imaging via Second-order Graph Matching (NeurIPS'23)
> > >
> > > ---
> > > ---
> > > We would like to sincerely thank you once again for your careful review and insightful suggestions. Your feedback has been extremely helpful to us to improve the paper. We are also grateful for your positive feedback on the novelty of our work. **We would greatly appreciate it if you could acknowledge our clarification**, and we sincerely hope that the responses above help resolve your concerns.
> > >
> > > **Best regards and have a nice day!**
> > >
> > > **4029 authors**

---

### Decision · Program_Chairs · 2026-04-30

**Decision:**

Accept (regular)

**Comment:**

This paper proposes M-IDoL, a self-supervised medical foundation model that uses information decomposition to reduce cross-modality ambiguity while preserving modality-specific structure. The discussion centered on the gap between the paper's idealized theory and the observed routing behavior, as well as on the fairness of comparisons against models trained with different pretraining data. The rebuttal added useful analysis on routing, stability, and domain-specific baselines, but some reviewers still wanted a cleaner mechanistic account of why the method works as well as it does. Decision: Accept. The paper is not without open questions, but the overall contribution and scope were strong enough to warrant acceptance.